# ARCHON: AN ARCHITECTURE SEARCH FRAMEWORK FOR INFERENCE-TIME TECHNIQUES

## ABSTRACT

Inference-time techniques are emerging as highly effective tools to enhance large language model (LLM) capabilities. However, best practices for developing systems that combine these techniques remain underdeveloped due to our limited understanding of the utility of individual inference-time techniques and the interactions between them. Additionally, efficiently and automatically searching the space of model choices, inference-time techniques, and their compositions is challenging due to the large design space. To address these challenges, we introduce ARCHON, a modular framework for selecting, combining, and stacking layers of inference-time techniques to construct optimized LLM systems for target benchmarks. Rather than relying on a single LLM called once, we leverage a diverse set of LLMs and inference-time techniques, creating *LLM systems greater than the sum of their parts*. ARCHON defines an extensible design space, encompassing techniques such as generation ensembling, repeated sampling, ranking, fusion, critiquing, verification, and unit testing. It transforms the problem of building LLM systems into a hyperparameter optimization objective. Given the available LLMs, inference-time techniques, and compute budget, ARCHON utilizes hyperparameter search techniques to discover optimized architectures for target benchmark(s). We evaluate ARCHON architectures across a range of instruction-following, reasoning, and coding benchmarks, including MT-Bench, Arena-Hard-Auto, AlpacaEval 2.0, MixEval, MixEval Hard, MATH, and CodeContests. ARCHON architectures outperform frontier models, such as GPT-4o and Claude 3.5 Sonnet, on these benchmarks, achieving an average accuracy increase of 15.1 percentage points by using all available LLMs.

## 1 INTRODUCTION

Inference-time techniques are gaining traction as effective methods for improving model capabilities. Examples include generation ensembling, ranking, and fusion, where models in the ensemble are queried in parallel, their responses are ranked, and the best ones are fused into a single, higher quality output, respectively (Jiang et al., 2023b; Wang et al., 2024). Other types of inference-time techniques are based on querying a single LLM successively (via repeated sampling) and using a voting strategy or unit tests to select the top generation (Brown et al., 2024; Chen et al., 2024; Li et al., 2024a). We divide these existing inference-time techniques into three categories: *generative*, meaning that new candidate responses are drawn from the models (e.g. generation ensembling and repeated sampling), *reductive*, meaning that the existing responses are aggregated or filtered to keep the top responses (e.g. fusion and ranking), or *comparative*, meaning they provide analysis of candidate responses (e.g. critiquing and unit testing), as shown in Table 2.

Recent work has made progress towards building robust *inference-time architectures*, which are systems composed of one or more large language models (LLMs) and inference-time techniques. Examples include Mixture-of-Agents (MoA) (Wang et al., 2024) and LLM-Blender (Jiang et al., 2023b), as well as single-model systems like LeanStar (Lin et al., 2024) and rStar (Deng et al., 2024). However, our experiments show that existing architectures, such as MoA, still suffer from lack of generalization and become significantly less effective beyond the task(s) they were developed on (see Section 4.2). We argue that designing effective and generalizable inference-time architectures requires:

- **Understanding the Utilities of Inference-Time Techniques**: Inference-time architectures typically delegate their additional inference budget towards more model sampling calls (Chen et al., 2024; Brown et al., 2024), which can be effective for math and coding tasks. Other tasks such as instruction-following and reasoning are shown to benefit from additional techniques, including ranking and fusion (Wang

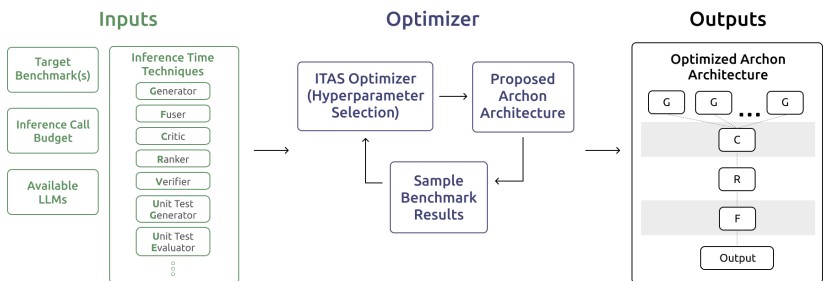

Figure 1: **Overview of ARCHON Framework**: Inference-Time Architecture Search (ITAS) requires the following inputs: target benchmarks, inference call budget, available LLMs, and available inference-time techniques (**left**). The ITAS algorithm uses Bayesian optimization (Snoek et al., 2012) (Section A.6) to select and test different ARCHON configurations (**middle**) before returning the optimized ARCHON architecture (**right**) for the target benchmarks (Section 3.3).

et al., 2024; Jiang et al., 2023b). While all of these methods are valuable, *it is essential to identify which inference-time techniques are most effective for different task categories.*

- **Understanding the Interactions Between Inference-Time Techniques**: While previous studies analyzed these techniques individually (e.g. generation sampling in Chen et al. (2024)), *we need a more comprehensive understanding of the relationships between different inference-time techniques* across different tasks (e.g. is it better to use more models or generate more samples per model?).
- **Efficiently and Automatically Searching the Large Design Space of Inference-Time Architectures**: Given a set of available LLMs and target tasks, there is currently no single prevailing inference-time architecture for maximizing downstream accuracy across all tasks (Table 1). The search space of inference-time architectures is expansive, requiring practitioners to make several key configuration decisions: *which LLMs to use, how many times to sample them, how to combine the candidate generations, what inference-time techniques to perform on the candidates, and more*. These motivate the need for adaptive and automated architecture search approaches.

In our work, we address each of these challenges. Firstly, we **evaluate the utilities of a comprehensive set of existing and proposed inference-time techniques** across instruction-following, reasoning, and coding tasks. Using both open-source and closed-source models, we examine a range of techniques such as *ensembling, fusion, ranking, critiquing, and verification* and introduce new methods such as *model-based unit test generation and evaluation* (Sections 3.1 and 3.2).

Secondly, we **analyze the interactions between inference-time techniques**, and explore the benefits of adding new models and new techniques individually. We find that candidate fusion substantially improves the quality of the final response generation, and when combined with additional techniques like critiquing, verifying, and ranking, can improve generation quality beyond the oracle best candidate from individual (non-fused) responses (Figure 3; Figure 7). Additionally, we find that candidate verification, unit test generation, and unit test evaluation are most effective for reasoning tasks, whereas critiquing and ranking are effective across instruction-following and reasoning tasks (Section 3.1; Table 12).

Thirdly, drawing upon our analysis of inference-time techniques, we present **ARCHON**, a framework for building inference-time architectures. ARCHON utilizes automatic **inference-time architecture search (ITAS)** algorithms to maximize generation quality for a wide range of tasks, including instruction-following, reasoning, and coding. Our ARCHON framework and ITAS algorithms draw inspiration from neural architectures and neural architecture search (NAS) (Zoph & Le, 2017; Ren et al., 2021; Liu et al., 2018; 2021), respectively. ARCHON is constructed of *layers of LLMs*, in which LLMs within the same layer run in parallel, but each layer runs sequentially. The layers perform different inference-time techniques, either transforming the number of candidate responses through generation and fusion (analogous to linear transformations) or reducing the number of candidate responses to improve quality (akin to non-linearities) (Section 3.1). The number of generators, samples per model, fusion layers, fusion models per layer, and more, are all treated as hyperparameters for optimization in our ITAS algorithms (Section 3.3).

Overall, our work makes the following contributions: **(1)** We develop ARCHON, an open-source modular framework for designing LLM systems that combine inference-time techniques (Section 3.1). We utilize ITAS as the optimizer engine for ARCHON, which enables automated inference-time architecture search for target

benchmarks, leveraging Bayesian optimization (Snoek et al., 2012; Nardi et al., 2019) (Section 3.3). ARCHON is plug-and-play, allowing users to select from existing inference-time techniques (or add new ones) and specify their desired objective functions to optimize for accuracy, latency, and cost. **(2)** We demonstrate increased performance as we scale up the layers of inference-time techniques and combine multiple approaches together, allowing us to discover effective new combinations of inference-time techniques (Sections 3.2, 4.2, A.2). We find that sequentially applying critique, ranking, top-k selection, and then fusion is a highly effective composition (Figure 3; Table 1), and we demonstrate the effectiveness of model-based unit test generation and evaluation for improving coding capability (Figure 5). **(3)** Our best ARCHON architectures surpass both single-call LLMs (e.g. GPT-4o and Claude-3.5 Sonnet) and prior top-performing inference-time architectures (e.g. Mixture-of-Agents (Wang et al., 2024)), boosting state-of-the-art performance by 15.1 percentage points, on average, across a diverse set of instruction-following, reasoning, and coding benchmarks (Table 1; Figure 5): MT-Bench, Arena-Hard-Auto, Alpaca-2.0 Eval, MixEval, MixEval Hard, MATH, and CodeContests (Zheng et al., 2023; Li et al., 2024b; 2023; Ni et al., 2024; Hendrycks et al., 2021; Li et al., 2022). Even when just using open-source LLMs, ARCHON architectures on average surpass single-call state-of-the-art (SOTA) LLMs by 11.2 percentage points.

## 2 RELATED WORK

**Scaling Laws of Language Models**: Language models (Touvron et al., 2023; Jiang et al., 2023a; Team et al., 2024a; OpenAI et al., 2024) have transformed the field of artificial intelligence across a vast number of domains and tasks. LLMs are pretrained on substantial amounts of textual data before being further aligned with human preferences through instruction fine-tuning (Wei et al., 2022; Chung et al., 2022), DPO (Rafailov et al., 2023), KTO (Ethayarajh et al., 2024), RLAIF (Bai et al., 2022b), and other techniques. As language models continue to gain improved abilities with further scaling of data, parameters, and compute (Kaplan et al., 2020; Gadre et al., 2024), the cost of developing new LLMs is ever increasing, requiring the curation of trillions of new tokens as well as substantial GPU-hours for pretraining. Furthermore, as the current state-of-the-art in LLMs are primarily closed-source APIs, such as OpenAI's GPT-4o (OpenAI et al., 2024), Google's Gemini (Team et al., 2024b) and Anthropic's Claude (Anthropic, 2024), it is difficult to effectively explore and push the frontier of existing LLMs without being able to manipulate the parameters of these closed-source models and employing techniques such as continual pretraining (Jin et al., 2021), instruction fine-tuning (Wei et al., 2022), data mixing (Ye et al., 2024), chain-of-thought (Wei et al., 2023), among others.

**Inference-Time Techniques**: Inference-time architectures combine multiple frozen LLMs and inference-time techniques (e.g., generation ensembling, sampling, ranking, and fusion), achieving superior performance compared to individual models. Notable works include Mixture-of-Agents (MoA) (Wang et al., 2024), LLM Blender (Jiang et al., 2023b), RouteLM (Ong et al., 2024), Smoothie (Guha et al.), and various approaches around *compound AI*, which are AI systems that use multiple components (e.g. LLMs, retrievers, tool use, APIs, etc.) (Chen et al., 2024; Davis et al., 2024; Lewis et al., 2020; Shao et al., 2024; Kapoor et al., 2024). LM frameworks like DSPy (Khattab et al., 2023) and TextGrad (Yuksekgonul et al., 2024) have emerged for orchestrating LMs and other components. Even with a single LLM, various techniques can improve performance by building better reasoning strategies, such as OpenAI's o1 (OpenAI, 2024b), Chain of Thought (Wei et al., 2023), and Branch-Solve-Merge (Saha et al., 2024), as well as inference-time frameworks, such as ADAS (Hu et al., 2024) and AFlow (Zhang et al., 2024).

Despite these advancements, challenges remain in developing inference-time architectures. Many architectures focus on additional generations (Jiang et al., 2023b; Chen et al., 2024; Davis et al., 2024), which is effective for reasoning tasks (Brown et al., 2024). However, for tasks like chat and instruction-following, techniques such as fusion and ranking are useful (Wang et al., 2024; Jiang et al., 2023b). For tasks without built-in verification, additional compute for reasoning and verification can improve accuracy (Davis et al., 2024). We still lack understanding of trade-offs between different inference-time techniques. Prior studies have explored limited aspects of configurations, often focusing on specific benchmarks (Jiang et al., 2023b; Wang et al., 2024; Chen et al., 2024; Li et al., 2024a). It's crucial to efficiently develop inference-time architectures, as optimal configurations vary based on benchmarks, available models, and inference call limits (Section 4.2). To address these challenges, we analyzed multiple inference-time techniques (Section 3.1) and developed the ARCHON framework for automating the development of inference-time architectures with ITAS (Section 3.3).

## 3 INFERENCE-TIME TECHNIQUES FOR ARCHON

With the proliferation of inference-time techniques, ARCHON introduces a simple framework that unifies different approaches, providing structure for understanding and combining various techniques. Our framework not only incorporates methods for generating, ranking, and fusing candidates inspired by previous work

(Wang et al., 2024; Jiang et al., 2023b) but also integrates new approaches for critiquing, verifying, and unit testing candidate responses.

Below, we elaborate on the structure, inputs, and outputs of each of the inference-time techniques, which we also include in Table 2. Then, we discuss how to combine the different techniques into an inference-time architecture (Section 3.2) and the relationships between the different inference-time techniques (Section A.2) before finally exploring automatic approaches for constructing inference-time architectures (Section 3.3).

## 3.1 LLM COMPONENTS OF ARCHON

In this section, we discuss the *LLM components* of ARCHON, which are LLMs that perform a specific inference-time technique. We test an array of different components inspired by recent work, incorporating approaches for generating, ranking, and fusing candidates (Wang et al., 2024; Jiang et al., 2023b) as well as approaches for improving candidate response quality through critiquing, verifying, and unit testing (Bai et al., 2022a; Zheng et al., 2023). The components and their prompts are summarized in Table 2 and Section A.1.

**Generator** is an LLM that takes in the instruction prompt and outputs candidate responses. Generators can be called in parallel to perform *generation ensembling* (i.e. calling multiple LLMs in parallel) (Wang et al., 2024), or sampled multiple times (Brown et al., 2024). When calling the Generators in parallel, you can sample one or more LLMs one or more times. The number of models, samples, and temperature for generation can be varied based on model configuration.

**Fuser** is an LLM that, given an instruction prompt and a set of proposed responses as input, combines these responses to generate one or more higher-quality fused responses that better address the instruction prompt.

**Ranker** is a LLM that, given an instruction prompt and a set of proposed responses as input, ranks the candidate generations based on their quality, producing a ranked list of responses as output.

**Critic** is an LLM that, given an instruction prompt and a set of proposed responses as input, produces a list of strengths and weaknesses for each response, which is then used to improve the quality of the final response (Section 3.2; Figure 3).

**Verifier** is a LLM that verifies whether a provided candidate response has appropriate reasoning for a given instruction prompt. It proceeds in two stages: **Stage #1** takes in the instruction prompt and a candidate response as input and outputs reasoning for why the candidate response is correct; **Stage #2** takes in the instruction prompt, candidate response, and produced reasoning before outputting reasoning and a verdict (i.e. binary [Correct] or [Incorrect]) for whether or not the candidate response is correct according to the provided instruction prompt and reasoning.

**Unit Test Generator** is a LLM that takes only the instruction prompt as input and outputs a list of unit tests that assess the accuracy and relevance of candidate responses. These unit tests are verified by the Unit Test Evaluator to rank different responses. Each test is a concise statement that can be passed or failed. We make the number of unit tests generated a configurable choice for the unit test generator but we find 5-10 generated unit tests to be most effective with our set of LM prompts (Section 4.2; Figure 5). For examples, please see Table 10.

**Unit Test Evaluator** is a LLM that takes in the instruction prompt, candidate response(s), and set of unit tests before outputting the candidate response(s), ranked in descending order by how many unit tests they pass. We use model-based unit test evaluation by prompting the LLM to provide reasoning and verdicts for each unit test across each of the candidate responses. By aggregating the unit test verdicts for each candidate response, the unit test evaluator ranks the candidate responses. For reasoning tasks, particularly coding tasks, it can be useful to compare different candidate responses by the number of unit tests they pass to gauge for quality (Figure 5).

## 3.2 COMBINING THE LLM COMPONENTS

**Overview**: Inspired by the structure of neural networks (Hinton et al., 1992), ARCHON is constructed of layers of LLM components (Figure 1; Section 3.1). Each layer is composed of sets of these LLM components that are called in parallel, performing a text-to-text operation to the instruction prompt and the candidate responses from the previous layer. Furthermore, like a neural network, some layers perform *transformations* of the provided list of strings (e.g. the Generator and Fuser components), converting a list of strings into a different list of strings (the numbers of candidates can vary from the original number of candidates). Other components introduce non-linearities into the ARCHON structure, performing filtering of the list of strings (e.g. Ranker and Verifier). Ultimately, the inputs and outputs for each layer is always a list of strings, whether that is the

instruction prompt (e.g. a list with a single string) or a list of candidate responses (e.g. a list of many strings). If a list of strings are outputted at the last layer of the ARCHON structure, the first string in the list is returned.

Unlike a classical neural network, no weights are learned between the LLM components and the layers; in turn the ARCHON architecture can be deployed off-the-shelf without any tuning. Additionally, a single state is transformed sequentially from the input layer to the final output; this single state is the initial instruction prompt and the current candidate responses. In Figure 2, we provide an example ARCHON architecture composed of six layers.

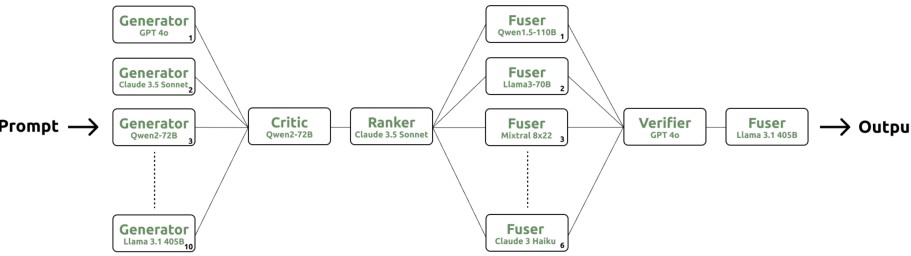

Figure 2: **Example ARCHON Architecture**: This architecture starts with ten generator models, followed by a critic model, a ranker model, one layer of six fuser models, a verifier model, and finishes with a fuser model.

**Rules for Construction**: The LLM components in Section 3.1 can only be placed in specific orders:

1. Only one type of module can be present in any given layer.
2. Generator components must and can only be placed in the first layer of ARCHON; you can put multiple Generators or a single Generator in the layer.
3. The Critic component must come before a Ranker or a Fuser, otherwise the generated strengths and weaknesses cannot be incorporated into generation ranking or fusion, respectively.
4. Ranker, Critic, Verifier, and Unit Test Generator/Evaluator layers can go anywhere in the ARCHON structure (besides the first layer); for each of these components, it must be the one and only module in its layer.
5. Fuser components can also be placed anywhere in the ARCHON structure (besides the first layer); you can put multiple Fusers or a single Fuser in the layer.
6. Unit Test Generators and Evaluators are placed in layers next to each other: generator first, then evaluator.

We provide an overview of the available placements and configurations for each LLM module in Table 3. We also analyze the different interactions between each LLM component and find increased ARCHON performance as we scale the "layers" of inference-time techniques by combining multiple approaches together sequentially (Section A.2).

**Performance Gains from Scaling Inference-Time Techniques**: By scaling both the layers of inference-time techniques and the diversity of inference-time techniques included, we were able to significantly improve AR-CHON performance across instruction-following, reasoning, and coding tasks (Figure 3). In particular, repeated model sampling and additional ensemble models led to substantial gains (Figure 4), leading to 9.3 and 18.5 percentage point increases, respectively. On coding tasks, additional samples provided the largest marginal benefits, leading to a 56% boost in Pass@1 for CodeContests when repeated sampling was combined with model-based unit test generation/evaluation (Figure 1). Multiple layers of Fusers were found to be particularly effective for instruction-following tasks, delivering notable performance improvements as the number of layers increased (Figure 3). In reasoning tasks, incorporating the Verifier and Unit Test Generator/Evaluator modules alongside the Fuser improved performance by filtering out flawed responses, contributing to significant performance gains in tasks like MixEval and CodeContests. For detailed analysis of interactions between LLM components, please see Section A.2, where we perform a series of ablation experiments in which we vary ARCHON component combinations (Table 12) and the models used in the combinations (Table 13; Table 14; Table 15; Table 16).

## 3.3 INFERENCE-TIME ARCHITECTURE SEARCH (ITAS)

In this section, we explore different approaches for finding the best inference-time architecture (for a given task) through *inference-time architecture search* (ITAS). Due to compute resources, we pre-filtered certain

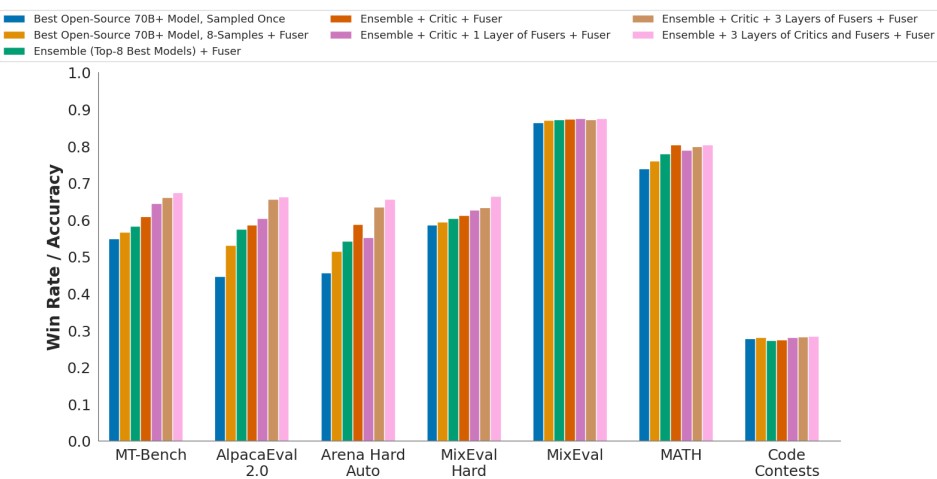

Figure 3: **Performance Gains from Scaling *Layers* of Inference-Time Techniques**: We generally observe performance improvements as we scale the critic and fusion layers. Compared to sampling the best open-source model once, our inference-time architecture with an 8-model ensemble, 3 layers of critic and fusion (8 models in each layer), and a final fusion performs on average 17.3% higher. For MixEval and CodeContests, we find that alternative inference-time architectures are more effective than generator ensembles and fusion layers. We break-down our results for MixEval and MixEval-Hard by subdataset in Section 4.2 (Table 31; Table 32). For CodeContests, we show the effectiveness of increased generator sampling combined with model-based unit test generation/evaluation in Figure 5. The results were calculated from 10 independent evaluation runs.

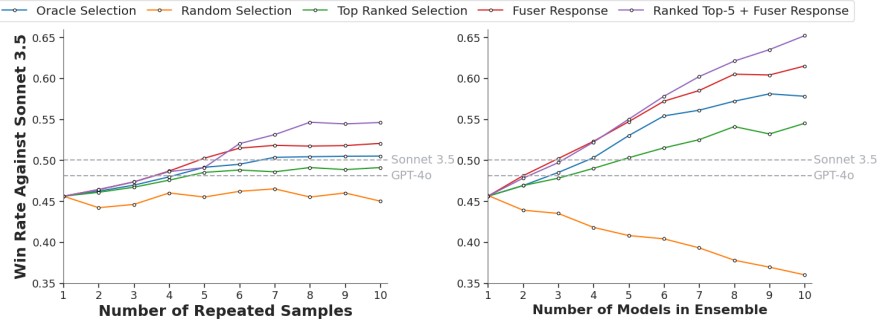

Figure 4: **Performance Gains from Repeated Sampling, Ensembling, Ranking, and Fusing on Arena-Hard-Auto**: The ARCHON win-rate continues to grow significantly as we scale model sampling **(left)** or add additional models to the generator ensemble **(right)**, increasing by 9.3% and 18.5%, respectively. These best results are achieved by selecting the top-5 responses and fusing them. The ensemble models are added based on their individual performance on this task, from best to worse (Table 18). The results were calculated from 10 independent evaluation runs.

ways of combining LLM components to reduce the search space while still building effective inference-time architectures. While it is possible to expand the search space of potential ARCHON architectures (e.g. different temperatures for generative LLM components, alternative prompts for each LLM component, multiple layers of Generator modules, additional LLM components for ARCHON, etc.), we use our analysis from Section 3.2 to selectively limit our search space to configurations that fit our rules for ARCHON: starts with a layer of Generator modules, followed by layers performing fusing, ranking, critiquing, verifying, and unit testing.

**Search Hyperparameters**: We selected five main axes for the hyperparameters in our search:

1. **Top-$K$ Generators for Ensemble**: The top-$K$ models to be used for the initial Generator ensemble, ranges from 1 to 10. The top-$K$ models are the best-$K$ LLMs for the given task, based on their individual performances (Table 18).

2. **Top-$K$ Generator Samples**: The number of samples gathered from each Generator in the ensemble (it is the same for all the models), ranges from 1 to 5. For Code-Contests, we explore high sample settings: [1, 10, 100, 500, 1000].
3. **Number of Fusion Layers**: Ranges from 1 to 4. The last fusion layer will always have a single Fuser.
4. **Top-$K$ Fusers**: Number of models used for each fusion layer, ranges from 2 to 10 and increases by 2 each time.
5. **Evaluation Layer**: Option to add a Verifier, Unit Test Generator/Evaluator, or neither before the last Fuser layer.

By combining all the hyperparameters, we create a search space of 18,750 configurations by multiplying each of the configuration option counts together ($10 * 5 * 5^{(4-1)} * 3 = 18,750$). However, we remove configurations that are not viable: configurations in which the number of initial generations exceeds the context window of the fusers (i.e. 24 candidate generations) and configurations with only one fuser layer but multiple fusers declared. This reduces our search space to 9,576 configurations. For these configurations, we add critic and ranker layers before each fuser layer since they've been shown to have added benefits across the benchmarks explored (Figure 7; Figure 3). The ranker selects the top-5 candidate generations to send to the next layer. The unit test generator uses a default setting of 5 unit tests generated.

**Search Methodology**: Within ITAS, we use *Bayesian Optimization* to select the most promising hyperparameter configurations (Snoek et al., 2012; Nardi et al., 2019). For generator ensemble, we add the models to the pool in a greedy manner, starting from the best performing model (on average) on the target benchmarks. For each fuser ensemble layer, we use the same approach, adding the best fuser models in a greedy manner. To rank them, we evaluate their fusion performance on the samples from an ensemble of top 10 generator models. We found that the best generator and fusion models could vary widely across datasets, making it beneficial to perform these rankings for new datasets (Table 18). For search, we use a 20% sample of each dataset for guiding architecture search to improve the evaluation speed while getting meaningful development signal.

Overall, Bayesian Optimization was the most effective search algorithm for constructing ARCHON systems, outperforming other methods like random and greedy search by more efficiently finding optimal configurations (Section A.6). It found the best architectures in 96.0% of iterations and required 88.5% fewer evaluations than greedy search and 90.4% fewer than random search (Figure 13). The effectiveness of Bayesian optimization increases with the number of initial testing points, up to around 230-240 samples, after which further testing is better focused on configuration search (Table 26). However, for limited inference call budgets (<20 calls), Bayesian optimization is less effective, and traditional methods like greedy search may perform comparably (Table 27).

For our implementation, we use a Python package of Bayes global optimization with Gaussian processes. As inputs, our Bayes implementation takes in the integer lists of configuration choices for the generators (i.e. number of models and samples), layers of fusers, numbers of fusers per layer, and final verifier / unit tester. Bayes algorithm then proceeds to select different combinations of integers from these lists in its search process, iteratively evaluating each generated ARCHON architecture on the development set to find the optimal ARCHON configuration. For more information, please see Section A.6.

## 4 EXPERIMENTS

Our experiments focus on four questions: **(1)** how does ARCHON compare to existing SOTA LLMs and multi-LLM systems? **(2)** how does ARCHON performance compare across tasks? **(3)** how does ARCHON performance compare when optimized for a set of tasks vs. an individual task? **(4)** what are ARCHON's current limitations and plans for future work?

### 4.1 BENCHMARKS AND MODELS

**Benchmarks**: We evaluate our models with several benchmarks for instruction-following, reasoning, and coding: MT-Bench (Zheng et al., 2023), AlpacaEval 2.0 (Li et al., 2023), Arena Hard Auto (Li et al., 2024b), MixEval (Ni et al., 2024), MixEval-Hard, MATH (Hendrycks et al., 2021), and CodeContests (Li et al., 2022). We provide an overview of each dataset in Table 29, where we compare their query counts, scoring type, evaluation metrics, reference models, and judge models. Since we perform ITAS on a randomly sampled 20% subset of each benchmark, we evaluate on the remaining held-out 80% subset of the benchmark (Table 1; Figure 5) (for ARCHON performances on the entire benchmarks, please see Table 28). The delta between the ARCHON performance on the entire benchmark vs. 80% held-out subset is relatively small:

| | Approaches | Infer. Calls | Input Tokens | Output Tokens | TFLOPs per Token | MT Bench W.R. | Alpaca Eval 2.0 L.C. W.R. | Arena Hard Auto W.R | MixEval Hard Acc. | MixEval Acc. | MATH Pass @1 |
|---|---|---|---|---|---|---|---|---|---|---|---|
| **LM** | GPT-4o | 1 | 95 | 549 | Unk. | 44.2% ±0.5 | 57.8% ±0.6 | 80.6% ±0.6 | 63.4% ±0.2 | 87.5% ±0.3 | 73.2% ±0.4 |
| | Claude 3.5 Sonnet | 1 | 105 | 602 | Unk. | N/A | 52.7% ±0.4 | 81.4% ±0.4 | 68.7% ±0.2 | 89.1% ±0.2 | 73.1% ±0.7 |
| | Llama 3.1 405B | 1 | 118 | 631 | 0.81 | 44.1% ±0.3 | 40.7% ±0.5 | 64.5% ±0.7 | 66.0% ±0.3 | 88.2% ±0.2 | 75.2% ±0.5 |
| **LM Systems** | MoA | 19 | 25,109 | 17,422 | 1.36 | 51.6% ±0.6 | 65.4% ±0.3 | 84.5% ±0.3 | 62.3% ±0.4 | 86.9% ±0.2 | 73.9% ±0.6 |
| | MoA Lite | 7 | 7,943 | 6,437 | 0.52 | 45.6% ±0.4 | 59.6% ±0.7 | 88.3% ±0.5 | 60.9% ±0.3 | 86.4% ±0.3 | 71.8% ±0.3 |
| | ADAS | 52 | 72,804 | 44,872 | Unk. | 66.3% ±0.7 | 60.1% ±0.5 | 85.4% ±0.4 | 64.2% ±0.2 | 87.0% ±0.2 | 74.5% ±0.8 |
| | AFlow | 48 | 68,596 | 41,748 | Unk. | 62.4% ±0.2 | 57.8% ±0.6 | 83.2% ±0.6 | 63.5% ±0.3 | 87.2% ±0.4 | 73.2% ±0.2 |
| | O1 Mini | Unk. | 112 | Unk. | Unk. | 57.1% ±0.3 | 57.8% ±0.4 | 79.3% ±0.8 | 70.8% ±0.2 | 87.0% ±0.3 | 81.7% ±0.4 |
| | O1 Preview | Unk. | 112 | Unk. | Unk. | 56.3% ±0.5 | 59.3% ±0.5 | 81.7% ±0.3 | 72.0% ±0.4 | 87.5% ±0.2 | 73.5% ±0.5 |
| **Open Src.** | General Purpose | 35 | 51,113 | 31,508 | 3.14 | 67.2% ±0.4 | 63.3% ±0.6 | 85.6% ±0.5 | 65.3% ±0.3 | 86.2% ±0.2 | 76.6% ±0.6 |
| | Task Specific | 44 | 63,157 | 39,949 | 3.71 | 71.1% ±0.6 | 67.1% ±0.4 | 89.6% ±0.4 | 67.5% ±0.2 | 88.8% ±0.3 | 81.9% ±0.3 |
| **Closed Src.** | General Purpose | 32 | 52,747 | 27,894 | Unk. | 72.7% ±0.3 | 63.9% ±0.7 | 86.2% ±0.7 | 67.5% ±0.4 | 87.2% ±0.2 | 77.9% ±0.7 |
| | Task Specific | 40 | 59,085 | 37,271 | Unk. | 77.0% ±0.5 | 68.9% ±0.5 | 90.5% ±0.3 | 72.6% ±0.3 | 89.5% ±0.3 | 81.6% ±0.4 |
| **All Src.** | General Purpose | 35 | 50,427 | 30,461 | Unk. | 76.2% ±0.7 | 66.4% ±0.3 | 89.8% ±0.6 | 69.8% ±0.2 | 87.3% ±0.4 | 79.3% ±0.5 |
| | Task Specific | 39 | 58,250 | 36,114 | Unk. | **79.5% ±0.4** | **69.0% ±0.6** | **92.5% ±0.5** | **72.7% ±0.3** | **89.7% ±0.2** | **82.1% ±0.6** |

Table 1: ARCHON*'s Strong Performance with ITAS Optimization on Open Source, Closed Source, and All Source Models*: Consistent outperformance over state-of-the-art LLMs across explored benchmarks. The standard error numbers were calculated from 10 independent evaluation runs.

only 0.44 percentage points, on average, across these datasets with an S.D. of 0.20 percentage points. For MixEval and MixEval Hard, we use the 2024-06-01 dataset release. For MT Bench, AlpacaEval 2.0, and Arena-Hard-Auto, the reference models are Claude 3.5 Sonnet, GPT-4-Turbo, and GPT-4-Turbo, respectively, while the judge models are GPT-4-0314, GPT-4-Turbo, and GPT-4-Turbo, respectively. For MATH, we evaluate a random sample of 200 problems from the dataset's test set. For CodeContests, we evaluate on the 140 test set questions that do not include image tags in the problem description.

**Models**: We test the efficacy of the ARCHON framework by creating various different ARCHON architectures (Section 4.4) across three model categories: 8B or less parameter models, 70B or more parameter models, and closed-source model APIs. For our 8B and 70B+ models, we selected the top-10 performing chat models for each parameter range on the Chatbot Arena Leaderboard (Chiang et al., 2024) as of July 2024. For our ARCHON architectures, we explore multiple model types: open-source, closed-source, and *all-source* (i.e. both open-source and closed-source available). For our closed-source model APIs, we include GPT-4o, GPT-4-Turbo, Claude Opus 3.0, Claude Haiku 3.0, and Claude Sonnet 3.5. We list and compare all of the models tested in the ARCHON framework in Table 17 and Table 18. For all the LLMs utilized and every ARCHON component, we set the generation temperature to 0.7. As baselines, we compare ARCHON against both SOTA LLMs (GPT-4o (OpenAI et al., 2024), Claude 3.5 Sonnet (Anthropic, 2024), and Llama 3.1 405B Instruct (AI@Meta, 2024)) as well as SOTA inference-time architectures (OpenAI's O1 (OpenAI, 2024a), MoA (Wang et al., 2024), ADAS (Hu et al., 2024), and AFlow (Zhang et al., 2024)).

## 4.2    ARCHON VS. CLOSED-SOURCE LLMS AND OTHER INFERENCE-TIME ARCHITECTURES

We start by comparing ARCHON architectures to existing SOTA closed-source LLMs and inference-time architectures across a set of instruction-following, reasoning, and coding tasks. Based on our results in Table 1, we find that ARCHON architectures consistently match or surpass existing approaches across all the benchmarks explored. ARCHON architectures with open-source models demonstrate a 11.2% average improvement over SOTA open-source approaches; for its worst performance, our open-source ARCHON architectures are only 3.6% above SOTA open-source approaches on AlpacaEval 2.0. ARCHON architectures with closed-source models achieve SOTA performance across MT Bench, Arena-Hard-Auto, MixEval, and MixEval-Hard, leading to a 15.8% average improvement over closed-source LMs and a 6.8% average improvement over open-source inference-time frameworks (i.e. MoA, ADAS, and AFlow). Furthermore, compared to these open-source inference-time frameworks, Archon is 20% more inference call efficient while having higher performances on all benchmarks tested. We also find that our best Archon architectures use 15.1% less input tokens and 13.5% less output tokens compared to the best alternative open-source inference-time frameworks. Compared to O1-preview and O1-mini, ARCHON's best targeted architectures beat them by 8.1% and 9.7%, on average, on MT Bench, AlpacaEval 2.0, Arena Hard Auto, MixEval, MixEval Hard, and MATH. On CodeContests,

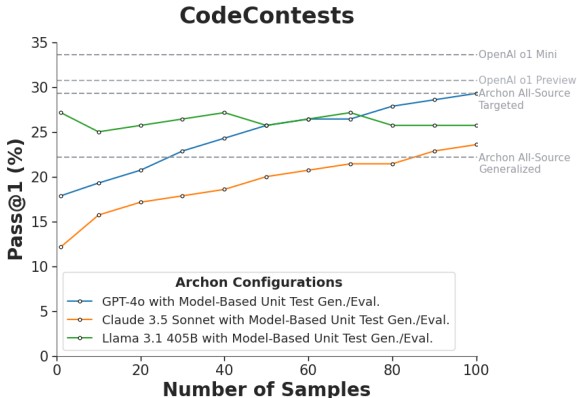

Figure 5: ARCHON *Performance Gains from Combining Multi-Sampling with LLM-based Unit-test Generation/Evaluation*: Strong performance improvements in Pass@1 as we scale the number of samples for GPT-4o and Claude 3.5 Sonnet. The standard error numbers were calculated from 10 independent evaluation runs.

O1-preview and O1-mini narrowly beats ARCHON by 1.7% and 5.3%, on average, as the O1 system is specially trained towards handling complex reasoning tasks like math and coding. Lastly, for approaches that use all models available, both open and closed-source, ARCHON achieves an average 10.9% improvement over existing SOTA single-call LLMs and an average 8.6% improvement over existing inference-time frameworks.

## 4.3 ARCHON BY TASK

We analyze ARCHON performance by task style: instruction-following tasks that use pairwise ranking for scoring, reasoning tasks that use accuracy-based metrics for scoring, and coding tasks that use Pass@1. On instruction-following tasks like MT Bench, AlpacaEval 2.0, and Arena-Hard-Auto, open-source ARCHON architectures outperform current open-source baselines by 10.0 percentage points, on average, while closed-source ARCHON outperforms current closed-source baselines by 20.1 percentage points (Table 1). On reasoning tasks like MixEval, MixEval-Hard, and MATH, open-source ARCHON architectures outperform existing open-source baselines by 2.9 percentage points while closed-source ARCHON architectures outperform current closed-baselines by 4.2 percentage points (Table 1). On coding tasks (i.e. CodeContests), open-source ARCHON architectures match existing open-source baselines (0.2 percentage points difference) and all-source ARCHON architectures outperform all-source baselines by 2.5 percentage points (Figure 5). All-source architectures of ARCHON outperform existing all-source baselines by 16.1 and 3.8 percentage points, on average, for instruction-following tasks and for reasoning tasks, respectively (Table 1).

**Instruction-Following and Reasoning**: With ARCHON, multiple models used for Generators and the depth of fusion layers lead to performance boosts on instruction-following tasks, increasing the richness of responses and allowing multiple iterations for step-by-step instruction-following (Table 19). For reasoning, while the performance boost from ARCHON is smaller when we consider the *aggregate* scores for MixEval and MixEval-Hard, we do see meaningful increases in performance when we create inference-time architectures for each individual task under MixEval and MixEval-Hard (Table 31; Table 32). When we create individual ARCHON architectures for each subtask, we see 3.7 and 8.9 percentage point increases in accuracy, on average, for MixEval and MixEval-Hard, respectively. This finding suggests that reasoning tasks (e.g. math, sciences, logic) require more individualized inference-time architectures for their particular queries.

**Coding**: We have observed that ensembling, fusion, and ranking techniques have limited impact on CodeContests (Figure 3). For example, when we apply the general all-source architecture from Table 29 to CodeContests problems, we achieve small gains from ARCHON (see Figure 5). One contributing factor is that, unlike the distribution of instruction-following/reasoning tasks, coding tasks tend to have one or two LLMs that perform substantially better than the rest of models (Table 18). However, when we add unit test generation/evaluation, and scale the number of samples, ARCHON's performance on CodeContests improves significantly (Figure 5), allowing us to boost GPT-4o Pass@1 performance by 56% for Pass@1 (from 25 to 41 out of 140 questions). For model-based unit test generation/evaluation, we generate 5 unit tests and use the LM to evaluate each candidate response against the generated unit tests, allowing us to rank the different candidate responses (details are provided in Section A.1)

## 4.4 TASK-SPECIFIC AND GENERAL-PURPOSE ARCHON ARCHITECTURES

**Task-Specific vs. General-Purpose**: We also compare custom ARCHON architectures, specifically configured to a single evaluation dataset ("Task-specific ARCHON Architectures"), and a generalized ARCHON architecture configured to handle all the evaluation datasets ("General-purpose ARCHON Architectures") (Table 1). For our three model selection settings for ARCHON (i.e. open-source, closed-source, and all-source), we utilize ITAS to find targeted ARCHON architectures for each task (7 architectures total) and find a single generalized ARCHON architecture for maximizing performance over all the tasks (Table 1). The benchmarks are concatenated together and shuffled for generalized Archon architecture search. For examples of targeted and generalized ARCHON architectures, please see Figure 2 and Section A.4.

We utilize ITAS to find the generalized ARCHON architectures in Table 1 (Section 3.3), maximizing performance over all of the benchmarks explored except CodeContests. While we use ITAS to find a targeted ARCHON architecture for CodeContests, we exclude the dataset from the generalized ARCHON architecture search since we found that ARCHON architectures for coding tasks are most effective with a different set of inference-time techniques compared to instruction-following and reasoning tasks (i.e. increased model sampling combined with model-based unit test generation/evaluation) (Section 3.2; Figure 3). For open-source models, we find that our generalized ARCHON architecture only lags behind the specialized ARCHON architectures by 3.4 percentage points, on average, across all the benchmarks, demonstrating the robustness of the ARCHON architecture found by the ITAS algorithms (Table 1). We see similar gaps between the generalized and specialized ARCHON architectures for closed-source models (4.0 percentage points) as well as the all-source models (3.3 percentage points) (Table 1).

**Insights from Architecture Construction**: We include examples of our learned effective generalized ARCHON architectures constructed by ITAS in Section A.4. For instruction-following and reasoning tasks, we found a generalizable ARCHON architecture to be most effective with multiple layers of critic-ranker-fuser, chained sequentially to improve candidate generation (Figure 9). However, the specific models chosen for these LLM components could change task by task, with some tasks benefiting from using a single SOTA closed-source LLM for all the components (e.g. Arena-Hard-Auto and MixEval) (Figure 11) whereas others benefited from a diversity of LLMs in their ensemble (e.g. MT Bench and MixEval-Hard) (Figure 9; Figure 10). Regardless of models used, we found that scaling inference layers including critics, rankers, and fusers improved performance on instruction-following and reasoning tasks (Figure 3; Section A.4). For instruction-following and reasoning tasks, the verifier module is more effective than the unit test generation/evaluation module for task-specific ARCHON architectures (Section 3.2; Table 12). For coding tasks, we found a high-sample setting to be the most effective, with added layers of unit test generation and evaluation to boost the quality of the final candidate generation (Figure 12; Figure 5).

## 4.5 LIMITATIONS AND FUTURE WORK OF ARCHON

**Parameter Count**: The ARCHON framework is most effective with LLM with about 70B parameters or more. When we utilize the ARCHON architecture with only 7B open-source models, we get a notable decrease in performance (Table 21). The best 7B ARCHON configurations lag behind single SOTA (and much larger) models by 15.7% on across all the benchmarks, on average; 7B models work well for ranking but are less effective for critic and fusion.

**Latency and Costs**: Since ARCHON architectures make multiple LLM API calls successively for different operations (e.g. ensembling, critiquing, ranking, etc.), it can often take 5x more time than a single LLM API call (Section A.4). Furthermore, it can require calling multiple API endpoints for a single query, leading to increased expenditures (Table 22; Table 23). Note that these increases in compute costs and latency translate to higher quality responses, and can be justified in many application domains, such as science, math, programming, and complex customer service issues. For tasks in which speed is most preferred, future work should explore how distillation strategies (Sreenivas et al., 2024) could be used to pack the aggregate knowledge of ARCHON architectures into a smaller LM.

**ARCHON Components**: While ARCHON is a modular framework, allowing the easy incorporation of new LLMs, new inference-time techniques, and even tool use, we only explore seven LLM inference time techniques in our work (Section 3.1). The addition of new techniques is a promising avenue for future research. Furthermore, while different queries can be best suited by different ARCHON architectures (Table 31; Table 32), the ITAS algorithm selects the best single architecture for the evaluation set queries combined. Future architecture search could focus on dynamic selection of ARCHON components, LLMs, and tools on a query-by-query basis.

## 5 REPRODUCIBILITY STATEMENT

For the ARCHON model and benchmark configurations, we included the related information in Sections 4.1, 4.2, and A.1. For performing Inference-Time Architecture Search (ITAS), we included the related information in Sections 3.3 and A.6. We also included our code in the submission supplementary materials.

# A APPENDIX

## A.1 ARCHON LLM COMPONENTS

| Inference-Time Technique | Definition | Input | Output | Inference Cost | Domains |
|---|---|---|---|---|---|
| Generator | Generates a candidate response from an instruction prompt | Instruction Prompt | Candidate Response(s) | 1 call per cand. | All Domains |
| Fuser | Merges multiple candidate responses into a single response | Instruction Prompt + Candidate Response(s) | Fused Candidate Response(s) | 1 call per cand. | All Domains |
| Critic | Generates strengths/weaknesses for each candidate response | Instruction Prompt + Candidate Response(s) | Candidate Response(s) Strengths/Weaknesses | 1 call | All Domains |
| Ranker | Returns top-K candidate responses | Instruction Prompt + Candidate Response(s) | Ranked Candidate Response(s) | 1 call | All Domains |
| Verifier | Returns the candidate responses with verified reasoning | Instruction Prompt + Candidate Response(s) | Verified Candidate Response(s) | 2 calls per cand. | Reasoning Tasks |
| Unit Test Generator | Generates unit tests to evaluate the candidate responses | Instruction Prompt | Instruction Prompt + Unit Tests | 1 call | Reasoning Tasks |
| Unit Test Evaluator | Uses generated unit tests to evaluate candidate response | Instruction Prompt + Unit Tests + Candidate Response(s) | Scored Candidate Response(s) | 1 call per cand. | Reasoning Tasks |

Table 2: **Overview of ARCHON's Inference-time Techniques**: Definitions, Inputs, Outputs, Costs, and Application Domains.

| Module | Initial Layer Placement | Placement after Initial Layer | >1 Module in Layer | Increase Candidate Responses | Decrease Candidate Responses |
|---|---|---|---|---|---|
| Generator | Yes | No | Yes | Yes | No |
| Fuser | No | Yes | Yes | Yes | Yes |
| Ranker | No | Yes | No | No | Yes |
| Critic | No | Yes | No | No | No |
| Verifier | No | Yes | No | No | Yes |
| Unit Test Generator | No | Yes | No | No | No |
| Unit Test Evaluator | No | Yes | No | No | No |

Table 3: **Rules of ARCHON Construction**: Allowed combinations of each LLM component from Section 3.1.

```
<instruction here>.
```

Table 4: **Generator Prompt**

```
You have been provided with a set of responses with their individual critiques of strengths/weaknesses from various open-source models
to the latest user query. Your task is to synthesize these responses into a single, high-quality response. It is crucial to critically evaluate
the information provided in these responses and their provided critiques of strengths/weaknesses, recognizing that some of it may be biased
or incorrect. Your response should not simply replicate the given answers but should offer a refined, accurate, and comprehensive reply
to the instruction. Ensure your response is well-structured, coherent, and adheres to the highest standards of accuracy and reliability.
Responses from models:
1. <response #1>
Critique: <critique #1>
2. <response #2>
Critique: <critique #2>
...
N. <response #N>
Critique: <critique #N>
<instruction here>
```

(a) With Critiques

```
You have been provided with a set of responses from various open-source models to the latest user query. Your task is to synthesize these
responses into a single, high-quality response. It is crucial to critically evaluate the information provided in these responses, recognizing
that some of it may be biased or incorrect. Your response should not simply replicate the given answers but should offer a refined, accurate,
and comprehensive reply to the instruction. Ensure your response is well-structured, coherent, and adheres to the highest standards of
accuracy and reliability.
1. <response #1>
2. <response #2>
...
N. <response #N>
<instruction here>
```

(b) Without Critiques

Table 5: **Fuser Prompt: Without and With Critiques**

```
I will provide you with N responses, each indicated by a numerical identifier []. Rank the responses based on their relevance to the instruction:
<instruction here>.
[1] <response #1>
[2] <response #2>
...
[N] <response #N>
Instruction: <instruction here>.
Rank the N responses above based on their relevance to the instruction. All the responses should be included and listed using identifiers, in descending
order of relevance to the instruction. The output format should be [] > [], e.g., [4] > [2]. Only respond with the ranking results, do not say
any word or explain.
```

Table 6: **Decoder-Based Ranking Prompt**

```
You are a helpful assistant. I will provide you with N responses, each indicated by a numerical identifier (e.g., [1], [2], etc.). Rank the responses based
on their relevance to the instruction: <instruction here>.
[1] <response #1>
[2] <response #2>
...
[N] <response #N>
Instruction: <instruction here>.
Evaluate the N responses above based on their relevance to the instruction. All the responses should be included and listed using identifiers. For each
response, start the critique with the numerical identifier (e.g., [1]) followed by the strengths and weaknesses. You must include both strengths and weaknesses,
even if there are more of one than the other. At the end of each response's analysis, include two new lines to separate the critiques. Do not include any preface
or text after the critiques. Do not include any references to previous critiques within a critique. Start with the analysis for the first response and end with
the analysis for the last response. All of the N responses should be included and evaluated using identifiers. Structure each response's analysis as follows:
Strengths:
- <strength #1>
- <strength #2>
- <strength #n>
Weaknesses:
- <weakness #1>
- <weakness #2>
- <weakness #n>
```

Table 7: **Critic Prompt**

I will provide you with a response indicated by the identifier 'Response'. Provide reasoning for why the response accurately and completely addresses the instruction: `<instruction here>`.
Response: `<response>`
Instruction: `<instruction here>`.
Provide the reasoning for the response above based on its relevance, completeness, and accuracy when compared to the instruction. Do not include any preface or text after the reasoning.

Table 8: **Verifier Prompt**

**Instruction Prompt:** Given the following query, generate a set of `N` unit tests that would evaluate the correctness of responses to this query.
- The unit tests should cover various aspects of the query and ensure comprehensive evaluation.
- Each unit test should be clearly stated and should include the expected outcome.
- The unit tests should be in the form of assertions that can be used to validate the correctness of responses to the query.
- The unit test should be formatted like 'The answer mentions...', 'The answer states...', 'The answer uses...', etc. followed by the expected outcome.
- Solely provide the unit tests for the question below. Do not provide any text before or after the list. Only output the unit tests as a list of strings (e.g., ['unit test #1', 'unit test #2', 'unit test #3']).
Query: `<instruction here>`

(a) With Unit Test Cap

**Instruction Prompt:** Given the following query, generate a set of unit tests that would evaluate the correctness of responses to this query.
- The unit tests should cover various aspects of the query and ensure comprehensive evaluation.
- Each unit test should be clearly stated and should include the expected outcome.
- The unit tests should be in the form of assertions that can be used to validate the correctness of responses to the query.
- The unit test should be formatted like 'The answer mentions...', 'The answer states...', 'The answer uses...', etc. followed by the expected outcome.
- Solely provide the unit tests for the question below. Do not provide any text before or after the list. Only output the unit tests as a list of strings (e.g., ['unit test #1', 'unit test #2', 'unit test #3']).
Query: `<instruction here>`

(b) Without Unit Test Cap

Table 9: **Unit Test Generator Prompt: With and Without Unit Test Cap**

**Instruction Prompt:** Compose an engaging travel blog post about a recent trip to Hawaii, highlighting cultural experiences and must-see attractions.
1. Unit Test #1: The blog post mentions at least two cultural experiences specific to Hawaii.
2. Unit Test #2: The blog post highlights at least three must-see attractions in Hawaii.
3. Unit Test #3: The tone of the blog post is engaging and uses descriptive language that would appeal to readers interested in travel.
4. Unit Test #4: The blog post includes factual information about Hawaii's culture, such as local customs, festivals, or historical facts.
5. Unit Test #5: The blog post contains a clear narrative structure, including an introduction, main body, and a conclusion.

(a) Instruction-Following Query

**Instruction Prompt:** Alice and Bob have two dice. They roll the dice together, note the sum of the two values shown, and repeat. For Alice to win, two consecutive turns (meaning, two consecutive sums) need to result in 7. For Bob to win, he needs to see an eight followed by a seven. Who do we expect to win this game?
1. Unit Test #1: The response correctly identifies the winning condition for Alice (two consecutive sums of 7).
2. Unit Test #2: The response correctly identifies the winning condition for Bob (a sum of 8 followed by a sum of 7).
3. Unit Test #3: The response explains the probability of achieving two consecutive 7s when rolling two dice.
4. Unit Test #4: The response explains the probability of achieving an 8 followed by a 7 when rolling two dice.
5. Unit Test #5: The response provides a conclusion on who is more likely to win based on the probability analysis.

(b) Reasoning Query

Table 10: **Unit Test Examples**

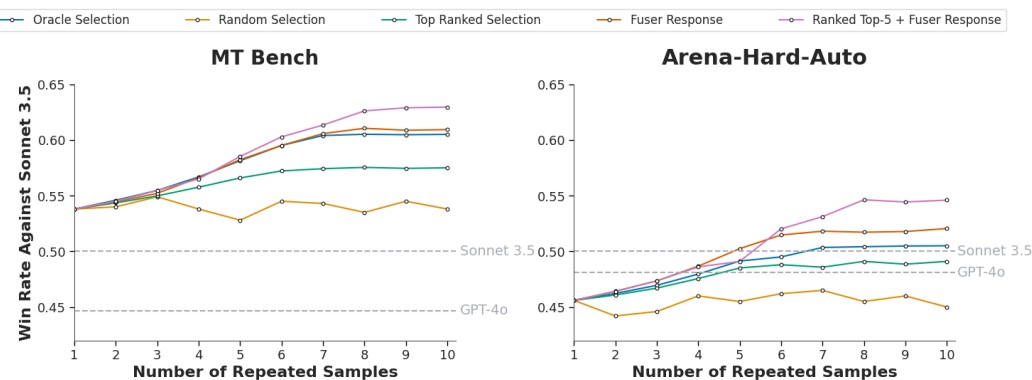

Figure 6: **Performance Gains from Applying Inference Time Techniques on a Single Model**: We repeatedly sample more responses for each individual query. For each sample count, we choose the best response in 5 different ways: **(1)** using an oracle (to get the upper bound for performance of best sample), **(2)** randomly, **(3)** using a ranker model, **(4)** by fusion, in which a model synthesizes a response based on all the samples, and **(5)** by ranking the top-5 best answers and then fusing them. For both MT Bench and Arena-Hard-Auto, we find that fusion is an effective technique. In particular, ranking the candidates first, and then selecting the top-5 and fusing them scores the highest. The best open-source model for these tasks across all the 70B+ models we are considering is WizardLM-2-8x22B (Xu et al., 2024) (see Table 18 for details). For both ranking and fusion, we use Qwen2 72B Instruct (Qwen, 2024).

Given the following query, candidate response, and unit tests, evaluate whether or not the response passes each unit test.
- In your evaluation, you should consider how the response aligns with the unit tests, retrieved documents, and query.
- Provide reasoning before you return your evaluation.
- At the end of your evaluation, you must finish with a list of verdicts corresponding to each unit test.
- You must include a verdict with one of these formatted options: '[Passed]' or '[Failed]'.
- Here is an example of the output format:
Unit Test #1: [Passed]
Unit Test #2: [Failed]
Unit Test #3: [Passed]
- Each verdict should be on a new line and correspond to the unit test in the same position.
- Here is the query, response, and unit tests for your evaluation:

Query: `<instruction here>`.

Candidate Response: `<response>`

Unit Tests:
```
Unit Test #1: <Unit Test #1>
Unit Test #2: <Unit Test #2>
...
Unit Test #N: <Unit Test #N>
```

Table 11: **Unit Test Evaluator Prompt**

## A.2 UTILITIES AND INTERACTIONS OF LLM COMPONENTS

In this subsection, we present our analysis of the effectiveness of each LLM component (i.e. the *Utility*) and the relationships between each component (i.e. the *Component Interactions*) by evaluating on *instruction-following tasks* (MT Bench, AlpacaEval 2.0, Arena-Hard-Auto), *reasoning tasks* (MixEval, MixEval-Hard, MATH) and *coding tasks* (CodeContests) (Section 4.1). For our ARCHON models, we utilize a host of 70B+ open-source models (Section 4.1; Table 17).

### A.2.1 GENERATOR

**Utility**: For our Generator module, we find additional model sampling to significantly boost performance (Figure 6), particularly for coding tasks (Table 1). In settings with a limited inference call budget, additional

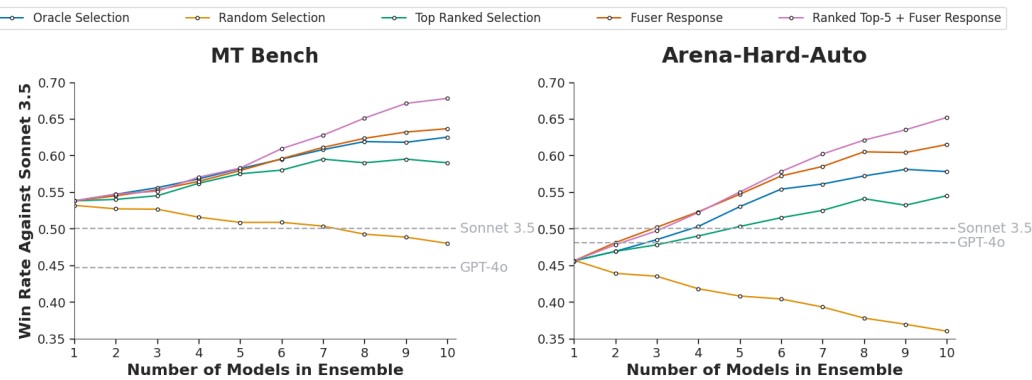

Figure 7: **Performance Gains from Applying Inference-Time Techniques on an Ensemble of Models**:
We incrementally add more models to the ensemble, which consists of open-source 70B+ models. The models
are added to the pool based on their performance for each task, from best to worse (see Table 18 for details).
For each ensemble size, we choose the best response in 5 different modes: **(1)** using an oracle (to get the upper
bound for performance of best individual response in the ensemble), **(2)** randomly, **(3)** using a ranker model, **(4)**
by fusion, in which one model synthesizes a response based on all the responses of the ensemble models, and **(5)**
ranking the top-5 best responses and then fusing them. For MT Bench and Arena-Hard-Auto, we find consistent
performance improvements as we add more models to the ensemble. We find that fusion is beneficial across
various ensemble sizes and in particular a fused candidate based on the top-5 ranked responses scores highest.
The ensemble approach scores higher than applying the same techniques on repeated samples from a single
best-performing model (see Figure 6). For both ranking and fusion, we use Qwen2 72B Instruct (Qwen, 2024).

model samples lead to the largest marginal benefit. We see a similar pattern for model ensembling, where
sampling from additional models leads to continual performance increases (assuming the models are ordered
from best to worst for the given task) (Figure 7).

### A.2.2 FUSER

**Utility**: For every benchmark explored, we found that the Fuser module substantially improved performance
(Figure 6; Figure 7; Figure 3). For the single-generation 10-model ensemble of 70B+ models, the Fuser
module improved downstream accuracy by 5.2 points, on average, compared to the single-generation best
model (Figure 7). When combined with the Ranker module for ranking the top-5 candidate responses, the
Fuser improved downstream accuracy by 7.3 points and 3.6 points, on average, compared to the single-sample
best model and the oracle best candidate response, respectively (Figure 7). Overall, we found that Fuser
efficacy increased as more candidate responses were provided, demonstrating that additional candidate
generations can continue to bolster inference-time architecture performance when combined with a Fuser.

In previous work like Mixture-of-Agents (MoA) (Wang et al., 2024), multiple layers of Fusers was found
to boost performance on some instruction-following tasks (i.e. MT Bench and Alpaca Eval 2.0). Across all the
benchmarks explored, we observed similar benefits in the ARCHON framework when adding multiple layers
of Fusers (Figure 3). However, based on our results in Figure 8, the number of Fuser layers needed to improve
performance varied by task, with some tasks receiving limited benefits from added layers (1-2 point increase
in accuracy for MixEval) while others experienced significant benefits with 3-4 fusion layers and more (2 to
5 point increase in win rate for MT Bench and Alpaca Eval 2.0). We attribute this distinction to the difference
in task requirements, with chat and instruction following tasks benefiting more from multiple iterations
of revisions through the multiple Fuser layers, leading to greater diversity in the final generation (Table 19).

**Component Interactions**: To better understand how the Fuser module works with the other LLM
components, we took the single-sample 10-model ensemble of Generators with a Fuser and tried adding each
of these components individually: a Critic, a Ranker, a Verifier, and a Unit Test Generator/Evaluator. Across
all of the benchmarks, the added candidate response analyses from the Critic improved the Fuser's ability
to effectively merge the different candidate responses, increasing performance by an average of 3.1 percentage
points (Figure 3). With the added Ranker, the ARCHON architecture improved the combined Ensemble
+ Critic + Fuser performance across all the benchmarks by 4.8 percentage points, on average (Figure 3).
The Ranker proved most effective for style-oriented tasks (e.g. MT Bench and AlpacaEval 2.0) since

the examples mostly focus on improving the instruction-guidance towards the provided prompt. With the added Verifier module (Figure 3), the performance of the Ensemble + Critic + Fuser configuration improved marginally for the instruction-following tasks (1.2 percentage points, on average, for MT Bench, AlpacaEval 2.0, and Arena-Hard-Auto). However, this configuration improved performance more on reasoning tasks (3.2 percentage points for MixEval and MixEval-Hard, on average), assisting generation by filtering out irrelevant or flawed answers before the final fusion step (Figure 3). The added Unit Test Generator and Evaluator was less effective for the instruction-following and reasoning tasks, only providing a 1.5 percentage points increase, on average, when added to the Ensemble + Critic + Fuser configuration (Table 12). However, for coding tasks, we found unit test generation and evaluation significantly improved performance, leading to a 10.7 percentage point increase (56% performance increase comparatively) as we scale model sampling (Table 1).

### A.2.3 CRITIC

**Utility**: The Critic module proved effective for every task we explored in Figure 3 and Table 12. With our 10-model 70B+ Generator ensemble and Fuser configuration of ARCHON, the added Critic improved performance on average by 3.1 percentage points across the benchmarks explored.

**Component Interactions**: While useful for most ARCHON architectures, the added strengths and weaknesses from the Critic module are particularly useful when combined with the Fuser module, helping guide generation fusion for a single layer and even useful when placed between multiple fusion layers (on average 3.2 percentage point boost across benchmarks in Figure 3). The Critic module was also effective with the Ranker module, providing additional information for comparing candidate responses (Figure 6) and leading to a 5.9 percentage point increase, on average (Table 12).

### A.2.4 RANKER

**Utility**: From our results in Table 12, Figure 6, and Figure 7, we found the Ranker to be most effective for instruction-following tasks, where pair-wise comparisons of answers focus on style and adherence to the prompt. To examine the candidate selection improvement provided by candidate ranking, we compare three approaches to the Ranker: (**1**) random selection of candidate generation, (**2**) oracle selection of candidate generation, and (**3**) the top-ranked candidate selected by our Ranker. For MT Bench and Arena-Hard-Auto, we find that the ranker improves generation output quality by 3.8% compared to random candidate selection and performs within 2.7% of oracle selection (Figure 6).

**Component Interactions**: Based on our benchmark results in Table 12, the Ranker pairs well with the Critic module; the provided strengths and weaknesses helps guide ranking, particularly for instruction-following tasks, improving performance by 5.9 percentage points, on average. Furthermore, the Ranker was also effective when paired with the Fuser; the filtered list of candidate responses helped improve the final condensed response produced by the Fuser by 3.8 percentage points, on average (Figure 7). When paired with the Verifier and Unit Test Generator, the Ranker had neutral effects; performances changed marginally, either positively or negatively by 1-2 percentage points (Table 12).

Overall, our findings demonstrate the value of added Rankers for instruction-following and reasoning tasks when paired with Fusers. We find that when Rankers are used alone with an ensemble of Generators, their performance lags behind the 10-sample best single model configuration by 3.0 percentage points, on average (Table 12). Additionally, our findings show the importance of building better rankers for more complex reasoning tasks, such as math and coding, which is a challenge also raised by Brown et al. (2024).

### A.2.5 VERIFIER

**Utility**: The Verifier was most effective for the reasoning benchmarks explored in Table 12. When just using a 70B+ Generator ensemble with Verifier module after generation, the ARCHON configuration lagged behind the ARCHON ensemble and fuser configuration by 1.5 percentage points, on average, across all benchmarks explored. This suggests that the Verifier is most effective when combined with other inference-time techniques.

**Component Interactions**: As noted in Section A.2.2, the Verifier augmented the performance of the Critic and Fuser on reasoning tasks (e.g. Arena-Hard-Auto, MixEval, MixEval-Hard), boosting performance by 3.7 percentage points, on average, when combined together with these modules. Overall, the Verifier is most powerful when augmenting additional components for tasks requiring verification of intermediate steps and the final response (Table 12). Therefore, the Verifier was less helpful for instruction-following tasks (e.g. MT Bench and AlpacaEval) but more effective for reasoning tasks (e.g. Arena-Hard-Auto and MixEval).

| | Model / LLM System | # of Infer. Calls | MT Bench W.R. | AlpacaEval 2.0 L.C. W.R. | AlpacaEval 2.0 Raw W.R. | Arena Hard Auto W.R. | MixEval Hard Acc. | MixEval Acc. | MATH Acc. | Code Contests Acc. |
|---|---|---|---|---|---|---|---|---|---|---|
| **Control** | Best Open-Source 70B+ Model, Sampled Once | 1 | 55.0% ±0.4 | 44.7% ±0.5 | 37.1% ±0.6 | 45.6% ±0.5 | 58.7% ±0.2 | 86.5% ±0.3 | 73.5% ±0.6 | **27.1% ±0.4** |
| | Ensemble + *Fuser* | 9 | 58.4% ±0.6 | 57.5% ±0.4 | 51.3% ±0.5 | 54.3% ±0.7 | 60.5% ±0.3 | 87.3% ±0.2 | 75.5% ±0.3 | 22.0% ±0.7 |
| | Ensemble + *Critic* + *Fuser* | 10 | 60.9% ±0.3 | 58.7% ±0.6 | 65.8% ±0.3 | 58.8% ±0.4 | 62.4% ±0.4 | 87.4% ±0.3 | 77.0% ±0.5 | 24.5% ±0.5 |
| **Ablations** | Ensemble + *Ranker* | 9 | 52.5% ±0.7 | 54.7% ±0.5 | 47.6% ±0.4 | 50.5% ±0.6 | 58.2% ±0.2 | 86.8% ±0.4 | 71.5% ±0.4 | 23.5% ±0.6 |
| | Ensemble + *Verifier* | 24 | 53.2% ±0.5 | 56.2% ±0.3 | 50.2% ±0.7 | 52.4% ±0.3 | 56.5% ±0.3 | 85.6% ±0.2 | 76.0% ±0.7 | 24.9% ±0.3 |
| | Ensemble + *Unit Test Gen./Eval.* | 18 | 51.5% ±0.4 | 54.4% ±0.6 | 49.4% ±0.5 | 46.1% ±0.8 | 55.2% ±0.4 | 86.0% ±0.3 | 75.0% ±0.5 | 25.1% ±0.4 |
| | Ensemble + *Ranker* + *Fuser* | 10 | 62.5% ±0.8 | 60.3% ±0.4 | 63.6% ±0.6 | 57.2% ±0.5 | 60.1% ±0.2 | 87.6% ±0.3 | 76.0% ±0.6 | 23.6% ±0.5 |
| | Ensemble + *Verifier* + *Fuser* | 25 | 60.5% ±0.3 | 59.4% ±0.7 | 58.7% ±0.3 | 59.2% ±0.4 | 65.1% ±0.3 | 87.5% ±0.2 | 78.0% ±0.4 | 24.5% ±0.7 |
| | Ensemble + *Unit Test Gen./Eval.* + *Fuser* | 17 | 61.4% ±0.6 | 58.5% ±0.5 | 55.1% ±0.4 | 56.4% ±0.7 | 62.8% ±0.4 | 86.9% ±0.3 | 77.0% ±0.8 | 26.3% ±0.6 |
| | Ensemble + *Critic* + *Verifier* + *Fuser* | 25 | 61.3% ±0.5 | 60.0% ±0.3 | 61.0% ±0.7 | 59.5% ±0.3 | 65.5% ±0.2 | 87.8% ±0.4 | 78.0% ±0.3 | 24.8% ±0.4 |
| | Ensemble + *Critic* + *Ranker* + *Fuser* | 11 | **64.7% ±0.4** | **62.6% ±0.6** | **72.4% ±0.5** | **60.9% ±0.6** | **67.0% ±0.3** | **88.3% ±0.2** | **79.5% ±0.5** | 24.1% ±0.5 |

Table 12: **Impact of Different Compositions of ARCHON's Inference-Time Techniques**: We see increased task performances from adding new LLM components to ARCHON. For CodeContests, we find that there is a single model (Llama 3.1 405B Instruct) that performs considerably better than the rest of the LLMs studied, making it more effective leverage additional model sampling (Table 1). For our ensemble, we use the best 8 open-source 70B+ models for the task (Table 18). For our fuser, critic, ranker, and verifier components, we use the best fuser model found for the task (Table 18). For each evaluation benchmark, we explain its configuration in Table 29 and Section 4.1. The standard error numbers were calculated from 10 independent evaluation runs.

| | Model / LLM System | # of Infer. Calls | MT Bench W.R. | AlpacaEval 2.0 L.C. W.R. | Arena Hard Auto W.R. | MixEval Hard Acc. | MixEval Acc. | MATH Acc. | Code Contests Acc. |
|---|---|---|---|---|---|---|---|---|---|
| **Control** | Single Generation | 1 | 44.2% ±0.6 | 57.8% ±0.5 | 48.1% ±0.7 | 63.4% ±0.3 | 87.5% ±0.2 | 73.2% ±0.4 | 17.9% ±0.3 |
| | Ensemble + *Fuser* | 11 | 53.7% ±0.3 | 59.5% ±0.6 | 49.7% ±0.5 | 65.5% ±0.2 | 82.0% ±0.3 | 70.7% ±0.6 | 16.0% ±0.4 |
| | Ensemble + *Critic* + *Fuser* | 12 | 56.1% ±0.7 | 59.7% ±0.4 | 53.9% ±0.6 | 67.4% ±0.4 | 82.0% ±0.2 | 71.8% ±0.5 | 18.9% ±0.6 |
| **Ablations** | Ensemble + *Ranker* | 11 | 47.6% ±0.4 | 49.7% ±0.5 | 45.5% ±0.4 | 63.3% ±0.3 | 81.6% ±0.4 | 66.5% ±0.7 | 17.9% ±0.5 |
| | Ensemble + *Verifier* | 11 | 48.4% ±0.5 | 51.2% ±0.7 | 47.7% ±0.8 | 61.4% ±0.2 | 80.5% ±0.3 | 71.0% ±0.3 | 23.0% ±0.4 |
| | Ensemble + *Unit Test Gen./Eval.* | 21 | 46.8% ±0.8 | 49.3% ±0.3 | 41.2% ±0.5 | 60.2% ±0.4 | 80.7% ±0.2 | 69.9% ±0.8 | 24.0% ±0.7 |
| | Ensemble + *Ranker* + *Fuser* | 12 | 58.0% ±0.2 | 60.1% ±0.6 | 52.2% ±0.3 | 65.0% ±0.3 | 82.0% ±0.2 | 71.0% ±0.4 | 18.0% ±0.3 |
| | Ensemble + *Verifier* + *Fuser* | 12 | 55.8% ±0.6 | 54.2% ±0.4 | 60.3% ±0.7 | 67.0% ±0.2 | 82.5% ±0.3 | 73.1% ±0.6 | 22.4% ±0.5 |
| | Ensemble + *Unit Test Gen./Eval.* + *Fuser* | 22 | 56.5% ±0.3 | 61.4% ±0.5 | 51.6% ±0.4 | 67.7% ±0.4 | 81.7% ±0.2 | 72.0% ±0.5 | **25.4% ±0.6** |
| | Ensemble + *Critic* + *Verifier* + *Fuser* | 13 | 56.6% ±0.7 | 62.0% ±0.3 | 55.0% ±0.6 | 68.5% ±0.3 | 82.7% ±0.4 | 73.5% ±0.3 | 22.2% ±0.4 |
| | Ensemble + *Critic* + *Ranker* + *Fuser* | 13 | **60.0% ±0.4** | **62.8% ±0.6** | **56.2% ±0.5** | **69.4% ±0.2** | **88.5% ±0.3** | **75.0% ±0.7** | 18.5% ±0.5 |

Table 13: **ARCHON Component Compositions with GPT-4o**: The ensemble uses generates 10 samples for the given query. The standard error numbers were calculated from 10 independent evaluation runs.

| | Model / LLM System | # of Infer. Calls | MT Bench W.R. | AlpacaEval 2.0 L.C. W.R. | Arena Hard Auto W.R. | MixEval Hard Acc. | MixEval Acc. | MATH Acc. | Code Contests Acc. |
|---|---|---|---|---|---|---|---|---|---|
| **Control** | Single Generation | 1 | 32.1% ±0.7 | 38.5% ±0.5 | 30.4% ±0.6 | 45.2% ±0.3 | 69.5% ±0.2 | 61.0% ±0.5 | 10.5% ±0.6 |
| | Ensemble + *Fuser* | 11 | 44.2% ±0.3 | 43.0% ±0.6 | 40.2% ±0.4 | 46.0% ±0.4 | 73.0% ±0.3 | 61.2% ±0.7 | 6.0% ±0.4 |
| | Ensemble + *Critic* + *Fuser* | 12 | 46.6% ±0.5 | 44.2% ±0.4 | 44.4% ±0.7 | 47.9% ±0.2 | 73.0% ±0.4 | 62.3% ±0.3 | 8.4% ±0.5 |
| **Ablations** | Ensemble + *Ranker* | 11 | 38.1% ±0.6 | 40.2% ±0.7 | 36.0% ±0.5 | 43.8% ±0.3 | 72.1% ±0.2 | 57.0% ±0.6 | 7.5% ±0.4 |
| | Ensemble + *Verifier* | 11 | 38.9% ±0.4 | 41.7% ±0.3 | 38.2% ±0.8 | 41.9% ±0.4 | 71.0% ±0.3 | 61.0% ±0.4 | 19.0% ±0.7 |
| | Ensemble + *Unit Test Gen./Eval.* | 21 | 37.3% ±0.8 | 39.8% ±0.6 | 31.7% ±0.3 | 40.7% ±0.2 | 71.2% ±0.4 | 60.4% ±0.8 | 22.0% ±0.3 |
| | Ensemble + *Ranker* + *Fuser* | 12 | 48.0% ±0.2 | 45.6% ±0.5 | 42.7% ±0.6 | 45.0% ±0.2 | 73.0% ±0.4 | 61.0% ±0.5 | 8.0% ±0.6 |
| | Ensemble + *Verifier* + *Fuser* | 12 | 46.3% ±0.5 | 44.7% ±0.4 | 45.0% ±0.4 | 50.5% ±0.4 | 73.0% ±0.3 | 63.6% ±0.3 | 18.6% ±0.5 |
| | Ensemble + *Unit Test Gen./Eval.* + *Fuser* | 22 | 47.0% ±0.3 | 43.9% ±0.7 | 42.1% ±0.7 | 48.2% ±0.2 | 72.2% ±0.4 | 62.0% ±0.6 | **23.5% ±0.4** |
| | Ensemble + *Critic* + *Verifier* + *Fuser* | 13 | 47.1% ±0.7 | 46.0% ±0.3 | 45.0% ±0.5 | 52.4% ±0.3 | 73.2% ±0.5 | 63.5% ±0.4 | 18.4% ±0.7 |
| | Ensemble + *Critic* + *Ranker* + *Fuser* | 13 | **50.5% ±0.4** | **48.3% ±0.6** | **46.7% ±0.3** | **55.1% ±0.4** | **73.7% ±0.3** | **65.0% ±0.5** | 8.1% ±0.5 |

Table 14: **ARCHON Component Compositions with GPT-4o-mini**: The ensemble uses generates 10 samples for the given query. The standard error numbers were calculated from 10 independent evaluation runs.

| | Model / LLM System | # of Infer. Calls | MT Bench
W.R. | AlpacaEval 2.0
L.C. W.R. | Arena Hard Auto
W.R. | MixEval Hard
Acc. | MixEval
Acc. | MATH
Acc. | Code Contests
Acc. |
|---|---|---|---|---|---|---|---|---|---|
| Control | Single Generation | 1 | N/A | 52.7% ±0.4 | 81.4% ±0.6 | 68.7% ±0.3 | 89.1% ±0.2 | 73.1% ±0.5 | 12.5% ±0.3 |
| Control | Ensemble + *Fuser* | 11 | N/A | 53.0% ±0.6 | 83.2% ±0.4 | 69.5% ±0.2 | 89.0% ±0.3 | 71.2% ±0.6 | 17.0% ±0.4 |
| Control | Ensemble + *Critic + Fuser* | 12 | N/A | 54.2% ±0.3 | 85.4% ±0.7 | 70.9% ±0.4 | 89.5% ±0.2 | 72.3% ±0.4 | 19.4% ±0.6 |
| Ablations | Ensemble + *Ranker* | 11 | N/A | 50.2% ±0.5 | 76.0% ±0.5 | 63.8% ±0.3 | 82.1% ±0.4 | 67.0% ±0.7 | 18.5% ±0.5 |
| Ablations | Ensemble + *Verifier* | 11 | N/A | 51.7% ±0.7 | 78.2% ±0.3 | 60.9% ±0.2 | 81.0% ±0.3 | 71.0% ±0.3 | 21.0% ±0.4 |
| Ablations | Ensemble + *Unit Test Gen./Eval.* | 21 | N/A | 49.8% ±0.4 | 71.7% ±0.8 | 58.7% ±0.4 | 81.2% ±0.2 | 70.4% ±0.8 | 22.0% ±0.7 |
| Ablations | Ensemble + *Ranker + Fuser* | 12 | N/A | 55.6% ±0.5 | 82.7% ±0.4 | 65.0% ±0.3 | 89.0% ±0.4 | 71.0% ±0.4 | 19.0% ±0.3 |
| Ablations | Ensemble + *Verifier + Fuser* | 12 | N/A | 54.7% ±0.3 | 85.0% ±0.6 | 70.5% ±0.2 | 89.3% ±0.3 | 73.6% ±0.6 | 21.6% ±0.5 |
| Ablations | Ensemble + *Unit Test Gen./Eval. + Fuser* | 22 | N/A | 53.9% ±0.6 | 82.1% ±0.5 | 68.2% ±0.4 | 89.2% ±0.2 | 72.0% ±0.5 | **23.5% ±0.6** |
| Ablations | Ensemble + *Critic + Verifier + Fuser* | 13 | N/A | 56.0% ±0.4 | 85.0% ±0.3 | 71.0% ±0.3 | 89.4% ±0.4 | 73.5% ±0.3 | 21.4% ±0.4 |
| Ablations | Ensemble + *Critic + Ranker + Fuser* | 13 | N/A | **58.3% ±0.5** | **86.7% ±0.7** | **73.0% ±0.2** | **89.7% ±0.3** | **75.0% ±0.7** | 19.1% ±0.5 |

Table 15: **ARCHON Component Compositions with Claude 3.5 Sonnet**: The ensemble uses generates 10 samples for the given query. The standard error numbers were calculated from 10 independent evaluation runs.

| | Model / LLM System | # of Infer. Calls | MT Bench
W.R. | AlpacaEval 2.0
L.C. W.R. | Arena Hard Auto
W.R. | MixEval Hard
Acc. | MixEval
Acc. | MATH
Acc. | Code Contests
Acc. |
|---|---|---|---|---|---|---|---|---|---|
| Control | Single Generation | 1 | 35.0% ±0.5 | 42.0% ±0.6 | 36.8% ±0.7 | 64.6% ±0.2 | 73.2% ±0.3 | 64.8% ±0.4 | 10.0% ±0.5 |
| Control | Ensemble + *Fuser* | 11 | 48.2% ±0.3 | 47.0% ±0.4 | 44.2% ±0.5 | 66.5% ±0.3 | 77.0% ±0.2 | 65.2% ±0.7 | 10.8% ±0.3 |
| Control | Ensemble + *Critic + Fuser* | 12 | 50.6% ±0.7 | 48.2% ±0.5 | 48.4% ±0.3 | 68.1% ±0.4 | 77.0% ±0.4 | 66.3% ±0.5 | 11.5% ±0.6 |
| Ablations | Ensemble + *Ranker* | 11 | 42.1% ±0.4 | 44.2% ±0.7 | 40.0% ±0.6 | 58.8% ±0.3 | 76.1% ±0.2 | 61.0% ±0.6 | 11.9% ±0.4 |
| Ablations | Ensemble + *Verifier* | 11 | 42.9% ±0.6 | 45.7% ±0.3 | 42.2% ±0.8 | 57.9% ±0.2 | 75.0% ±0.3 | 65.0% ±0.4 | 12.0% ±0.7 |
| Ablations | Ensemble + *Unit Test Gen./Eval.* | 21 | 41.3% ±0.8 | 43.8% ±0.6 | 35.7% ±0.4 | 55.7% ±0.4 | 75.2% ±0.2 | 64.4% ±0.8 | 13.0% ±0.3 |
| Ablations | Ensemble + *Ranker + Fuser* | 12 | 52.0% ±0.2 | 49.6% ±0.5 | 46.7% ±0.7 | 60.0% ±0.3 | 77.0% ±0.4 | 65.0% ±0.5 | 12.0% ±0.6 |
| Ablations | Ensemble + *Verifier + Fuser* | 12 | 50.3% ±0.5 | 48.7% ±0.4 | 48.7% ±0.5 | 67.5% ±0.2 | 77.0% ±0.3 | 67.6% ±0.3 | 10.5% ±0.5 |
| Ablations | Ensemble + *Unit Test Gen./Eval. + Fuser* | 22 | 51.0% ±0.3 | 47.9% ±0.7 | 46.1% ±0.6 | 64.2% ±0.4 | 76.2% ±0.2 | 66.0% ±0.6 | **14.3% ±0.4** |
| Ablations | Ensemble + *Critic + Verifier + Fuser* | 13 | 51.1% ±0.7 | 50.0% ±0.3 | 49.0% ±0.4 | 68.0% ±0.3 | 77.2% ±0.4 | 67.5% ±0.3 | 10.0% ±0.7 |
| Ablations | Ensemble + *Critic + Ranker + Fuser* | 13 | **54.5% ±0.4** | 52.3% ±0.6 | 50.7% ±0.3 | 70.4% ±0.2 | 77.7% ±0.3 | 69.0% ±0.5 | 11.5% ±0.5 |

Table 16: **ARCHON Component Compositions with Claude-3-Haiku**: The ensemble uses generates 10 samples for the given query. The standard error numbers were calculated from 10 independent evaluation runs.

### A.2.6 UNIT TEST GENERATOR AND EVALUATOR

**Utility**: The Unit Test Generator and Evaluator were most effective on reasoning and coding tasks, improving performance on benchmarks that required more verification steps, such as Arena-Hard-Auto, MixEval, MixEval-Hard, MATH, and CodeContests (Table 12). For the reasoning tasks, we found the unit test generator and evaluator to be most effective when combined with other components. When the 70B+ ensemble of Generators was only combined with unit tests, it was less effective for reasoning tasks like Arena-Hard-Auto and MixEval, lagging behind the ensemble and fuser configuration by 3.1 percentage points. This inspired us to look into other inference-time techniques combinations for unit test generation, such as increased sampling and fusion. When we increased generation sampling and added unit test generation/evaluation for CodeContests, we see a 56% boost in Pass@1 performance (Table 1), increasing from 17.9 to 29.3 Pass@1.

**Component Interactions**: When combined with the Fuser module, the Unit Test Generator and Evaluator improved performance by 2.1 percentage points across the benchmarks explored (Table 12). The combined ensemble, Unit Test Generator/Evaluator, and Fuser ARCHON configuration was most effective on the reasoning benchmarks, leading to a 2.5 percentage point boost, on average. For coding, the unit test generator and evaluator was most effective when combined with the best performing Generator (using large sample counts) and a final Fuser (subsection 4.2).

| Models | MT Bench | | Alpaca Eval 2.0 | | Arena Hard Auto | | MixEval | | MixEval Hard | | MATH | | CodeContests | |
|---|---|---|---|---|---|---|---|---|---|---|---|---|---|---|
| | Gen | Fusion | Gen | Fusion | Gen | Fusion | Gen | Fusion | Gen | Fusion | Gen | Fusion | Gen | Fusion |
| GPT-4o | 44.7% | 61.9% | **57.5%** | 64.5% | **48.1%** | 69.2% | 88.0% | **89.4%** | 63.6% | 65.4% | 72.0% | 75.5% | 17.9% | 19.4% |
| GPT-4-Turbo | 42.2% | **63.1%** | 55.0% | **65.8%** | **48.1%** | 61.9% | 88.9% | 89.0% | 64.1% | 64.4% | 74.5% | 76.5% | 9.3% | 14.2% |
| Claude 3 Opus | 30.9% | 57.2% | 40.5% | N/A | 27.0% | 47.9% | 88.3% | 88.2% | 63.6% | 64.0% | 72.5% | 71.0% | 10.0% | 12.5% |
| Claude 3.5 Sonnet | N/A | 71.9% | 52.37% | 63.6% | N/A | **73.2%** | **89.7%** | 89.3% | **68.9%** | **69.5%** | 72.0% | 74.5% | 12.1% | 15.5% |
| Qwen 2 72B Instruct | 35.0% | 59.7% | 37.48% | 56.0% | 14.5% | 49.5% | 86.5% | 87.5% | 58.7% | 61.1% | **76.0%** | **78.5%** | 3.6% | 5.2% |
| DeepSeek LLM 67B Instruct | 18.4% | 20.0% | 17.8% | 17.1% | N/A | N/A | 79.2% | N/A | 42.5% | N/A | 45.0% | N/A | 5.7% | N/A |
| Qwen 1.5 72B Chat | 24.7% | 46.3% | 36.6% | 55.7% | 14.4% | 36.4% | 84.5% | 82.1% | 50.3% | 52.2% | 62.5% | 65.5% | 15.0% | 13.9% |
| Qwen 1.5 110B Chat | 34.4% | 50.3% | 43.6% | 55.9% | 21.9% | 39.7% | 85.3% | 86.5% | 51.8% | 55.6% | 67.0% | 72.5% | 3.6% | 7.8% |
| Wizard 8x22B | **53.8%** | 57.2% | 44.7% | 50.6% | 45.6% | 51.2% | 83% | 78.1% | 54.3% | 50.4% | 69.0% | 58.5% | 7.1% | 10.4% |
| Llama 3.1 8B Instruct | 33.1% | 45.9% | 25.6% | 34.9% | 11.9% | 28.6% | 75.0% | 57.5% | 41.3% | 46.5% | 59.0% | 60.5% | 8.6% | 7.8% |
| Llama 3.1 70B Instruct | 45.0% | 51.9% | 35.6% | 40.2% | 23.8% | 37.2% | 85.7% | 83.5% | 61.1% | 65.5% | 69.0% | 71.5% | 20.7% | **23.4%** |
| Llama 3.1 405B Instruct | 44.7% | N/A | 40.3% | N/A | 28.4% | N/A | 88.9% | N/A | 66.2% | N/A | 74.5% | N/A | **27.1%** | N/A |

Table 18: **ARCHON Generation and Fusion Performances for Single Models**: For Alpaca Eval 2.0, we use the length-controlled win rate (LC WR). For fusion, we gather one candidate from each of the top-10 generator models.

## A.3   ARCHON LLM ANALYSIS

| Model | Source Code | Parameter Count | Max Sequence Length |
|---|---|---|---|
| GPT-4o (OpenAI et al., 2024) | Closed-Source | — | 128K |
| GPT-4-Turbo (OpenAI et al., 2024) | Closed-Source | — | 128K |
| Claude-3-Opus (Anthropic, 2024) | Closed-Source | — | 200K |
| Claude-3.5-Sonnet (Anthropic, 2024) | Closed-Source | — | 200K |
| Claude-3-Haiku (Anthropic, 2024) | Closed-Source | — | 200K |
| Llama-3.1-70B-Instruct (Dubey et al., 2024) | Open-Source | 70B | 8k |
| Llama-3.1-405B-Instruct (Dubey et al., 2024) | Open-Source | 70B | 8k |
| DeepSeek LLM 67B Chat (Guo et al., 2024) | Open-Source | 67B | 32k |
| Qwen2 72B Instruct (Qwen, 2024) | Open-Source | 72B | 32k |
| Qwen1.5 110B Chat (Bai et al., 2023) | Open-Source | 110B | 32k |
| Qwen1.5 72B Chat (Bai et al., 2023) | Open-Source | 72B | 32k |
| Mixtral 8x22B v0.1 (Jiang et al., 2024) | Open-Source | 176B | 32k |
| WizardLM 8x22B (Xu et al., 2024) | Open-Source | 176B | 32k |
| dbrx-instruct (Databricks, 2024) | Open-Source | 132B | 32k |
| princeton-nlp/Llama-3-Instruct-8B-SimPO (Meng et al., 2024) | Open-Source | 8B | 8k |
| princeton-nlp/Llama-3-Instruct-8B-DPO (Meng et al., 2024) | Open-Source | 8B | 8k |
| princeton-nlp/Llama-3-Instruct-8B-RDPO (Meng et al., 2024) | Open-Source | 8B | 8k |
| princeton-nlp/Llama-3-Instruct-8B-IPO (Meng et al., 2024) | Open-Source | 8B | 8k |
| Llama-3.1-8B-Instruct (Dubey et al., 2024) | Open-Source | 8B | 8k |
| Qwen2-7B-Instruct (Qwen, 2024) | Open-Source | 7B | 32k |
| Qwen/Qwen1.5-7B-Chat (Bai et al., 2023) | Open-Source | 7B | 32k |
| mistralai/Mistral-7B-Instruct-v0.2 (Jiang et al., 2023a) | Open-Source | 7B | 32k |
| cognitivecomputations/dolphin-2.2.1-mistral-7b (Hartford, 2024) | Open-Source | 7B | 32k |
| microsoft/Phi-3-mini-4k-instruct (Abdin et al., 2024) | Open-Source | 4B | 4k |
| HuggingFaceH4/zephyr-7b-beta (Tunstall et al., 2023) | Open-Source | 7B | 32k |
| microsoft/Phi-3-small-8k-instruct (Abdin et al., 2024) | Open-Source | 7B | 8k |
| snorkelai/Snorkel-Mistral-PairRM-DPO (Tran et al., 2023) | Open-Source | 7B | 32k |
| mistralai/Mistral-7B-Instruct-v0.3 (Jiang et al., 2023a) | Open-Source | 7B | 32k |

Table 17: **Models Tested with ARCHON**.

| | Jaccard Similarity (%) | | | | | | |
|---|---|---|---|---|---|---|---|
| Inference-Time Architecture | MT Bench | AlpacaEval 2.0 | Arena-Hard Auto | MixEval | MixEval Hard | MATH | Code Contests |
| Best Open-Source 70B+ Model, Sampled 8 Times + Fuser | 45.3% | 52.1% | 48.4% | 55.2% | 58.9% | 65.2% | 63.7% |
| Ensemble (8 Top Models), Sampled Once Each + Fuser | 31.6% | 34.1% | 28.9% | 38.6% | 40.9% | 57.1% | 53.4% |

Table 19: **Jaccard Similarities between Candidates Responses and Fused Response by Benchmark**: For the fuser, we use the best-performing 70B+ model for benchmark.

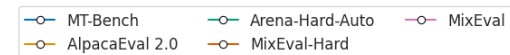

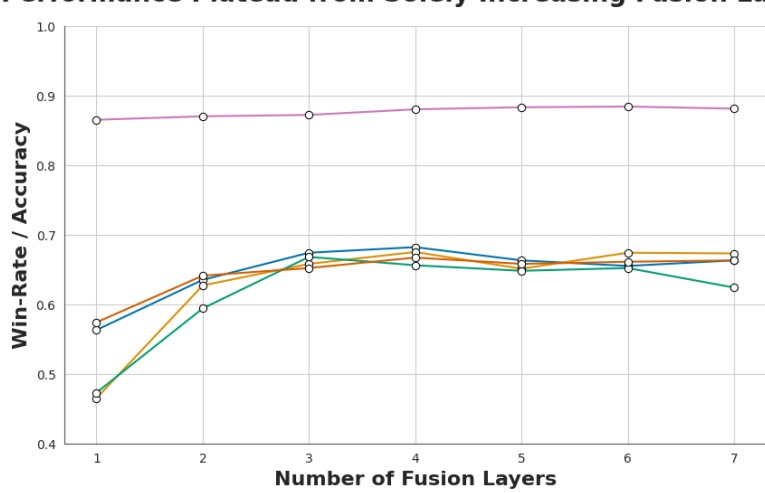

Figure 8: **Fusion Layer Efficacy by Benchmark**: From solely scaling the fusion layers, we see limited benefits across the benchmarks explored but when we add other inference-time techniques, such as Critic and Ranker, we see increased downstream performance as we continue scaling inference-time compute (Figure 3). We use an 8-model ensemble of the top Generator models for each benchmark (Table 18). For our Fuser layers, we use the best Fuser model for the final fuser layer (Table 18). For the intermediate layers, we use the top-8 Fuser models for each benchmark.

## A.4 ARCHON ARCHITECTURES

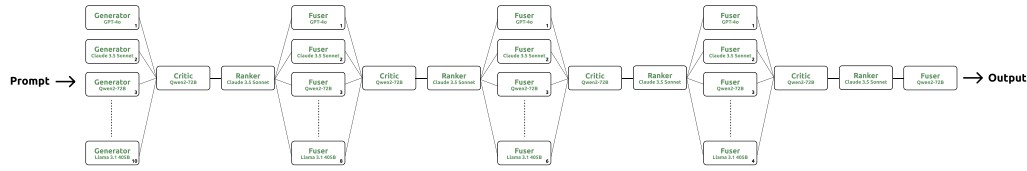

Figure 9: **All-Source Generalizable ARCHON Architecture**: Using ITAS, we found this all-source ARCHON configuration to be effective across the benchmarks explored (except for CodeContests). In the diagram above, we use 10 SOTA all-source LLMs to create multiple successive layers of critic, ranker, and fusers, with each successive fuser layer having less fusers to produce a "funneling" effect as the candidate generations are processed. The layers of critic, ranker, and fuser led to better candidate generations through iterative critique and rewriting. Each of the initial Generator models were sampled once.

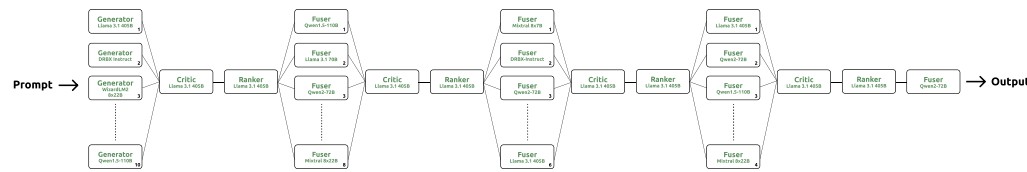

Figure 10: **Open-Source Generalizable ARCHON Architecture**: Using ITAS, we found this open-source ARCHON configuration to be effective across the benchmarks explored (except for CodeContests). In the diagram above, we use 10 SOTA open-source LLMs to create multiple successive layers of critic, ranker, and fusers, with each successive fuser layer having less fusers to produce a "funneling" effect as the candidate generations are processed. The layers of critic, ranker, and fuser led to better candidate generations through iterative critique and rewriting. Each of the initial Generator models were sampled once.

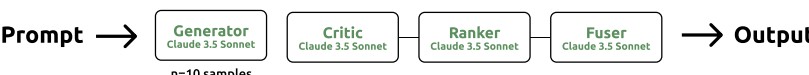

Figure 11: **All-Source ARCHON Architecture for Instruction-Following**: Using ITAS, we found Claude-3.5-Sonnet as a generator, critic, ranker, and fuser to be an effective targeted architecture for instruction-following tasks, such as MT Bench and AlpacaEval 2.0. The ranker picks the top-5 candidate responses to send to the fuser. Each of the initial Generator models were sampled once.

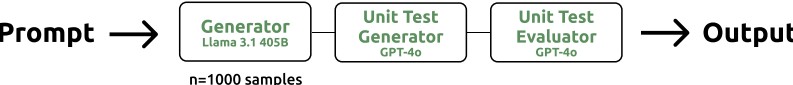

Figure 12: **All-Source ARCHON Architecture for CodeContests**: Using ITAS, we were able to get improved code generation on CodeConetsts by utilizing Llama 3.1 405B for generation and GPT-4o for unit testing (Figure 5). The unit test generator produces 10 unit tests for evaluation. Each of the initial Generator models were sampled once.

## A.5  ARCHON BY INFERENCE COMPUTE BUDGET, MODEL SIZE, AND COST

| | Number of Inference Calls | MT Bench | Alpaca Eval 2.0 | Arena Hard Auto | MixEval | MixEval Hard |
|---|---|---|---|---|---|---|
| **70B+ Models** | 1 | 55.0% | 44.7% | 45.6% | 86.5% | 61.1% |
| | 10 | 52.5% | 50.6% | 45.6% | 86.5% | 63.9% |
| | 20 | 65.3% | 60.4% | 59.4% | 89.0% | 65.0% |
| | 30 | 69.2% | 64.5% | **69.0%** | **89.5%** | **67.5%** |
| | 40 | 69.5% | **66.7%** | **69.0%** | **89.5%** | **67.5%** |
| | 50 | **71.6%** | **66.7%** | **69.0%** | **89.5%** | **67.5%** |
| **Closed Models** | 1 | 45.0% | 57.5% | 48.1% | 88.9% | 68.9% |
| | 10 | 57.1% | 63.2% | 68.4% | 90.0% | 70.1% |
| | 20 | 59.4% | 66.5% | 75.5% | **90.6%** | 70.5% |
| | 30 | 70.2% | **68.8%** | **77.4%** | **90.6%** | **72.9%** |
| | 40 | 75.5% | **68.8%** | **77.4%** | **90.6%** | **72.9%** |
| | 50 | **80.4%** | **68.8%** | **77.4%** | **90.6%** | **72.9%** |

Table 20: **ARCHON with Different Inference Budgets**: For AlpacaEval 2.0, we use the length-controlled win rate (LC WR).

| Models / LLM Systems | Datasets | | | | |
|---|---|---|---|---|---|
| | MT Bench | Alpaca Eval 2.0 | Arena Hard Auto | MixEval | MixEval Hard |
| SOTA Single-Model | **44.7%** | **57.5%** | **48.1%** | 68.9% | **89.7%** |
| Best Model, 1-Sample | 15.7% | 41.0% | 18.3% | 76.2% | 46.1% |
| Best Model - 10-Sample + Ranking | 16.5% | 43.2% | 18.9% | 78.4% | 48.5% |
| 10-Model, 1-Sample Ensemble + Ranking | 22.4% | 48.2% | 25.6% | **81.5%** | 52.9% |
| 10-Model, 1-Sample Ensemble + Fusion | 14.3% | 39.4% | 17.5% | 73.2% | 45.2% |
| 10-Model, 1-Sample Ensemble + Top-5 Ranking + Fusion | 15.9% | 41.2% | 18.0% | 75.1% | 46.9% |
| 10-Model, 1-Sample Ensemble + Critic + Fusion | 10.5% | 38.4% | 16.5% | 71.4% | 42.5% |

Table 21: **ARCHON with 7B Open-Source Models**: For AlpacaEval 2.0, we use the length-controlled win rate (LC WR). We use open-source 7B models for testing from Table 17.

| Models | Cost ($) per Million Input Tokens | Cost ($) per Million Output Tokens |
|---|---|---|
| Claude 3.5 Sonnet | $3 | $15 |
| Claude 3.0 Opus | $15 | $75 |
| GPT-4o | $5 | $15 |
| GPT-4-Turbo | $10 | $30 |
| TogetherAI - Llama 3.1 405B Instruct | $5 | $5 |
| TogetherAI - Llama 3.1 70B Instruct | $0.88 | $0.88 |
| TogetherAI - Other Models | $0.90 | $0.90 |

Table 22: **Model API Costs as of August 2024**

| Model / LLM System | Cost ($) per Query for Benchmark | | | | | | |
|---|---|---|---|---|---|---|---|
| | MT Bench | AlpacaEval 2.0 | Arena-Hard Auto | MixEval | MixEval Hard | MATH | Code Contests |
| Claude 3.5 Sonnet | 0.0305 | 0.0171 | 0.0212 | 0.0231 | 0.0226 | 0.0325 | 0.384 |
| GPT-4o | 0.0481 | 0.0236 | 0.0324 | 0.0357 | 0.0361 | 0.514 | 0.562 |
| Llama 3.1 405B Instruct | 0.0281 | 0.0174 | 0.0185 | 0.0212 | 0.0205 | 0.305 | 0.372 |
| General Purpose ARCHON Architecture | 0.364 | 0.189 | 0.195 | 0.284 | 0.252 | 0.375 | 0.461 |
| Task Specific ARCHON Architecture | 0.401 | 0.210 | 0.221 | 0.295 | 0.265 | 0.425 | 0.448 |

Table 23: **ARCHON Costs per Query by Benchmark**

## A.6 BAYESIAN OPTIMIZATION

Bayesian Optimization is a sequential design strategy for global optimization of black-box functions that are expensive to evaluate Snoek et al. (2012). It is particularly useful when dealing with functions that have unknown forms and are costly to evaluate, such as hyperparameter tuning in machine learning.

### A.6.1 OVERVIEW OF BAYESIAN OPTIMIZATION

The core idea behind Bayesian Optimization is to build a probabilistic model of the objective function and use it to select the most promising points to evaluate next. This process involves two main components:

1. **Surrogate Model**: A probabilistic model (often a Gaussian Process) that approximates the unknown objective function.

2. **Acquisition Function**: A function that guides the search for the optimum by suggesting the next point to evaluate, based on the surrogate model.

### A.6.2 STEPS IN BAYESIAN OPTIMIZATION

1. **Initialization**: Begin with a set of initial points $\mathcal{D} = \{(\mathbf{x}_1, y_1), (\mathbf{x}_2, y_2), ..., (\mathbf{x}_n, y_n)\}$, where $\mathbf{x}_i$ is the input, and $y_i = f(\mathbf{x}_i)$ is the objective function value at $\mathbf{x}_i$.
2. **Model Building**: Fit a surrogate model (e.g., Gaussian Process) to the observed data $\mathcal{D}$.
3. **Acquisition**: Use the acquisition function to select the next point $\mathbf{x}_{n+1}$ to evaluate:

$$\mathbf{x}_{n+1} = \operatorname*{argmax}_{\mathbf{x}} a(\mathbf{x} \mid \mathcal{D})$$

   where $a(\mathbf{x} \mid \mathcal{D})$ is the acquisition function.
4. **Evaluation**: Evaluate the objective function at $\mathbf{x}_{n+1}$ to get $y_{n+1} = f(\mathbf{x}_{n+1})$.
5. **Update**: Add the new data point $(\mathbf{x}_{n+1}, y_{n+1})$ to the dataset $\mathcal{D}$.
6. **Repeat**: Repeat steps 2-5 until convergence or a stopping criterion is met (e.g., budget exhausted, no significant improvement).

### A.6.3 GAUSSIAN PROCESS AS A SURROGATE MODEL

A Gaussian Process (GP) is commonly used as a surrogate model in Bayesian Optimization. It is defined by a mean function $\mu(\mathbf{x})$ and a covariance function (kernel) $k(\mathbf{x}, \mathbf{x}')$:

$$f(\mathbf{x}) \sim \mathcal{GP}(\mu(\mathbf{x}), k(\mathbf{x}, \mathbf{x}'))$$

Given a set of observations $\mathcal{D}$, the GP provides a predictive distribution for the objective function at a new point $\mathbf{x}$:

• **Predictive Mean**: The expected value of the function at $\mathbf{x}$:

$$\mu(\mathbf{x} \mid \mathcal{D}) = \mathbf{k}_n(\mathbf{x})^T \mathbf{K}_n^{-1} \mathbf{y}$$

   where $\mathbf{k}_n(\mathbf{x})$ is the covariance vector between $\mathbf{x}$ and the training points, and $\mathbf{K}_n$ is the covariance matrix of the training points.
• **Predictive Variance**: The uncertainty in the function value at $\mathbf{x}$:

$$\sigma^2(\mathbf{x} \mid \mathcal{D}) = k(\mathbf{x}, \mathbf{x}) - \mathbf{k}_n(\mathbf{x})^T \mathbf{K}_n^{-1} \mathbf{k}_n(\mathbf{x})$$

### A.6.4 ACQUISITION FUNCTIONS

Acquisition functions guide the search for the optimum by balancing exploration (trying out areas with high uncertainty) and exploitation (focusing on areas with high predicted values). Common acquisition functions include:

1. **Expected Improvement (EI)**:

$$\text{EI}(\mathbf{x}) = \mathbb{E}[\max(0, f(\mathbf{x}) - f(\mathbf{x}^+))]$$

   where $f(\mathbf{x}^+)$ is the best observed value so far.

2. **Probability of Improvement (PI)**:

$$PI(\mathbf{x}) = \mathbb{P}(f(\mathbf{x}) > f(\mathbf{x}^+) + \xi)$$

where $\xi$ is a small positive number.

3. **Upper Confidence Bound (UCB)**:

$$UCB(\mathbf{x}) = \mu(\mathbf{x}|\mathcal{D}) + \kappa\sigma(\mathbf{x}|\mathcal{D})$$

where $\kappa$ controls the trade-off between exploration and exploitation.

### A.6.5 SUMMARY OF BAYESIAN OPTIMIZATION

Bayesian Optimization iteratively uses a surrogate model to approximate the objective function and an acquisition function to decide where to sample next. By focusing on promising areas of the search space and systematically exploring uncertain regions, it efficiently optimizes complex, expensive-to-evaluate functions.

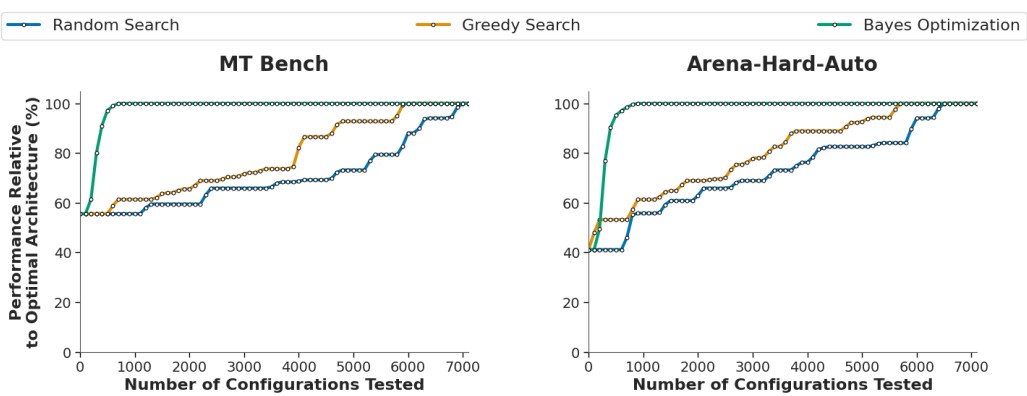

Figure 13: **Impact of Different Optimization Algorithms on Inference-Time Architecture Search (ITAS)**: On the benchmarks MT Bench and Arena-Hard-Auto, we compare four approaches for finding the optimal inference-time architecture: random search, greedy search, and Bayes Optimization. Bayes Optimization finds the optimal architecture in 88.5% less iterations compared to greedy search and 90.4% less iterations compared to random search.

### A.7 BAYES OPTIMIZATION VS. ALTERNATIVE APPROACHES

**Search Techniques**: Within the hyperparameter space, we explored three search algorithms for automating the development of inference-time architectures:

1. **Random Search**: Randomly selects a combination of hyperparameters for our ARCHON architecture.
2. **Greedy Search**: Starting with a base ARCHON configuration, marginally changes each hyperparameter and test if it improves performance or not. If it does, incorporate the change. If not, move on to the next hyperparameter.
3. **Bayesian Optimization**: Efficiently selects the most promising hyperparameter configurations for ARCHON by building a probabilistic surrogate model and leveraging an acquisition function for hyperparameter selection (Snoek et al., 2012; Nardi et al., 2019) (Section A.6).

To get our model ranking for the benchmark, we calculate the model ranking by testing each model individually on a 20% sample of each dataset benchmark in the first stage of the search. To get our fusion model ranking for the benchmark, we use the same approach, testing each model's fusion performance with an ensemble of 10 randomly selected models from the available set. From our experiments, we found that the best generator and fusion models could vary widely dataset to dataset, making it beneficial to perform these rankings for new datasets (Table 18). For search, we use the same 20% sample of each dataset that was

used for evaluating generation and fusion, allowing us to guide architecture search with improved evaluation speed while getting meaningful development signal.

**Comparing Search Algorithms**: In Figure 13, we compare the effectiveness of each search algorithm on our explored benchmarks. While random search guarantees the optimal ARCHON configuration, we found Bayesian optimization to be most effective in terms of tradeoff between finding the optimal configurations and minimizing the number of configurations tested. For 96.0% percent of the search iterations tested in Figure 13, we found that Bayesian optimization had the optimal configuration amongst the four explored search algorithms. We use 230 initial samples for our Bayes Optimization architecture search (Section A.6). Bayesian optimization also found the best architecture configuration in 88.5% less evaluations than greedy search and 90.4% less evaluations than random search.

**Bayesian Optimization Analysis**: In Table 26, we explore how the number of initial testing points, the number of exploration iterations, and the ARCHON inference call budget impacts the effectiveness of Bayesian optimization. Additional initial testing points continue improving search efficacy up until 230-240 samples, where testing would be better delegated towards configuration search. For lower inference call budgets with ARCHON (e.g. <20 inference calls), Bayesian optimization proved less effective, performing more similarly to greedy search or random search given the limited search space (Table 27). Therefore, Bayesian optimization is more effective for more open-ended ITAS with larger inference call budgets (e.g. >20 inference calls) whereas traditional component engineering might be better for more limited inference call budgets.

## A.8 ITAS Algorithms Comparisons

| # of Init. Points | % of Total Configs | Iter. till Max. Config. | Comb. Iter. |
|---|---|---|---|
| 200 | 2.18% | 353 | 553 |
| 210 | 2.29% | 324 | 534 |
| 220 | 2.40% | 301 | 521 |
| 230 | 2.51% | 284 | 514 |
| 240 | 2.61% | **261** | **501** |
| 250 | 2.72% | 265 | 515 |
| 260 | 2.83% | 256 | 516 |
| 270 | 2.94% | 252 | 522 |

Table 24: MT Bench

| # of Init. Points | % of Total Configs | Iter. till Max. Config. | Comb. Iter. |
|---|---|---|---|
| 200 | 2.18% | 478 | 678 |
| 210 | 2.29% | 431 | 641 |
| 220 | 2.40% | 415 | 635 |
| 230 | 2.51% | **382** | **612** |
| 240 | 2.61% | 389 | 629 |
| 250 | 2.72% | 385 | 635 |
| 260 | 2.83% | 372 | 632 |
| 270 | 2.94% | 368 | 638 |

Table 25: Arena-Hard-Auto

Table 26: **Bayesian Optimization Hyperparameter Comparisons**: On MT Bench and Arena-Hard-Auto, we compare Bayesian optimization configurations for the number of initial sample points. We find that 230 to 240 initial sample points minimizes the combined number of iterations (both initial sampling and exploring) to find the optimal configuration. For the configurations explored, the total number of hyperparameter choices is 9,576.

| | Iterations to Convergence | | | | |
|---|---|---|---|---|---|
| **Inference Budget** | 10 | 20 | 30 | 40 | 50 |
| Random Selection | 387 | 1152 | 2731 | 4359 | 5843 |
| Greedy Search | 343 | 984 | 2153 | 3045 | 4895 |
| Bayes Optimization | **254** | **386** | **452** | **515** | **589** |

Table 27: **ITAS Algorithms Comparison by Inference Call Budget**: For our comparison, we evaluate on MT Bench.

## A.9 ARCHON BENCHMARKS AND RESULTS

| | | Datasets | | | | | | | |
|---|---|---|---|---|---|---|---|---|---|
| | | MT Bench | Alpaca Eval 2.0 | | Arena Hard Auto | Arena Hard Auto | MixEval Hard | MixEval | MATH* |
| **Judge Model** | | GPT-4 0314 | GPT-4 Turbo | | GPT-4 Turbo | GPT-4 Turbo | N/A | N/A | N/A |
| **Reference Model** | | Claude 3.5 Sonnet | GPT-4 Turbo | | Claude 3.5 Sonnet | GPT-4 Turbo | N/A | N/A | N/A |
| **Model / LLM System** | **Infer. Calls** | W.R. | L.C. W.R. | Raw W.R. | W.R. | W.R | Acc. | Acc. | Pass @1 |
| GPT-4o - 2024-05-13 | 1 | 44.7% | 57.5% | 51.3% | 48.1% | 80.3% | 63.6% | 88.0% | 72.0% |
| Claude 3.5 Sonnet | 1 | N/A | 52.4% | 40.6% | N/A | 80.9% | 68.9% | 89.7% | 72.0% |
| Llama 3.1 405B Instruct | 1 | 44.7% | 40.3% | 37.7% | 28.4% | 64.1% | 66.2% | 88.9% | 74.0% |
| MoA | 19 | 51.6% | 65.1% | 59.8% | 52.2% | 84.2% | 62.5% | 87.3% | 72.5% |
| MoA Lite | 7 | 45.6% | 59.3% | 57.0% | 40.6% | 87.8% | 61.1% | 87.1% | 70.5% |
| **Open Source** — General-purpose ARCHON Architecture | 35 | 67.5% | 63.0% | 68.3% | 66.2% | 85.1% | 65.5% | 86.9% | 75.5% |
| **Open Source** — Task-specific ARCHON Architectures | 44 | 71.6% | 66.7% | 70.7% | 69.0% | 89.5% | 67.5% | 89.6% | **80.5%** |
| **Closed Source** — General-purpose ARCHON Architecture | 32 | 73.1% | 63.5% | 69.1% | 70.5% | 85.8% | 67.7% | 88.2% | 77.0% |
| **Closed Source** — Task-specific ARCHON Architectures | 40 | 77.5% | 68.4% | 72.1% | 74.4% | 90.2% | **72.9%** | 90.4% | 79.0% |
| **All Source** — General-purpose ARCHON Architecture | 35 | 76.8% | 65.8% | 70.2% | 72.5% | 89.3% | 70.1% | 88.1% | 78.0% |
| **All Source** — Task-specific ARCHON Architectures | 39 | **80.4%** | **67.6%** | **73.3%** | **76.1%** | **92.1%** | **72.9%** | **90.6%** | **80.5%** |

Table 28: **ARCHON's Strong Performance on the Complete Evaluation Datasets after ITAS Optimization**: We find that ARCHON's inference-time architectures consistently outperform single-call state-of-the-art LLMs, both open-source and closed-source baselines, when evaluating on the complete benchmarks (Table 29). We explore two configurations: ITAS for building custom ARCHON configurations for each individual benchmark and ITAS for building a single general-purpose ARCHON configuration for all the benchmarks (Section 4.4). We find that a general ARCHON configuration lags behind the custom ones by only 3.2 percentage points, on average, across our all-source settings, which suggests the efficacy of general-purpose inference-time architectures created with our framework. For Arena-Hard-Auto, we also include a configuration with Claude 3.5 Sonnet as a stronger reference model for comparison against ARCHON inference-time architectures and to mitigate bias from GPT judges towards GPT generations. For MT Bench, we use a GPT-4-0314 judge model instead of newer LLM judges to be consistent with previous results on this benchmark. For our task-specific ARCHON architectures, we also provide the average inference calls across the given benchmarks. For our full-list of models explored, please see Table 17. For MATH, we use a randomly sampled subset of size 200 for evaluation (Section 4.1; Table 29). We include our ARCHON architecture results on the held-out 80% subset of each evaluation benchmark in Table 1.

| Benchmark | Example Count | Reference Model | Judge Model | Scoring Type | Metric |
|---|---|---|---|---|---|
| AlpacaEval 2.0 | 805 | GPT-4-Turbo | GPT-4-Turbo | Pairwise Comparison | L.C. & Raw Win Rates |
| Arena-Hard-Auto | 500 | Claude-3.5-Sonnet GPT-4-0314 | GPT-4-Turbo | Pairwise Comparison | Win Rate |
| MT-Bench | 80 | Claude-3.5-Sonnet | GPT-4-0314 | Pairwise Comparison | Adjusted Win Rate |
| MixEval | 2000 | N/A | N/A | Ground Truth | Accuracy |
| MixEval-Hard | 500 | N/A | N/A | Ground Truth | Accuracy |
| MATH | 200 (sampled from 5000) | N/A | N/A | Ground Truth | Pass@1 |
| CodeContests | 140 (non-visual queries) | N/A | N/A | Ground Truth | Pass@1 |

Table 29: **Benchmark Overview**: Evaluation configurations for AlpacaEval 2.0 (Li et al., 2023), Arena-Hard-Auto (Li et al., 2024b), MT-Bench (Zheng et al., 2023), MixEval (Ni et al., 2024), MixEval Hard, MATH (Hendrycks et al., 2021), and CodeContests (Li et al., 2022)
.

| | Model / LLM System | Arena-Hard-Auto | |
|---|---|---|---|
| | | Score | C.I. |
| | Claude 3.5 Sonnet | N/A | N/A |
| | GPT-4o | 48.1% | (-2.3, 1.8) |
| | Llama 3.1 405B Instruct | 28.4% | (-2.7, 2.5) |
| **Open Source** | General-purpose ARCHON Architecture | 66.2% | (-2.4, 2.2) |
| | Task-specific ARCHON Architectures | 69.0% | (-2.8, 2.5) |
| **Closed Source** | General-purpose ARCHON Architecture | 70.5% | (-2.5, 2.0) |
| | Task-specific ARCHON Architectures | 74.4% | (-2.3, 1.6) |
| **All Source** | General-purpose ARCHON Architecture | 72.5% | (-2.5, 1.8) |
| | Task-specific ARCHON Architectures | **76.1%** | (-1.8, 2.2) |

Table 30: **ARCHON Results on Arena-Hard-Auto Results with Claude-3.5-Sonnet as Baseline Model**: The baseline model is Claude-3.5-Sonnet (default baseline model: GPT-4-0314) while the judge model is GPT-4-Turbo.

| Model / LLM System | Infer. Calls | MixEval - Sub-Datasets | | | | | | |
|---|---|---|---|---|---|---|---|---|
| | | GSM8K | TriviaQA | DROP | MATH | BBH | AGIEval | Average |
| GPT-4o - 2024-05-13 | 1 | 94.9 | 89.1 | 88.2 | **98.5** | 98.3 | 71.5 | 90.3 |
| Claude 3.5 Sonnet | 1 | 98.0 | 92.0 | 92.6 | 96 | 95.6 | 78.0 | 92.0 |
| Llama 3.1 405B Instruct | 1 | **98.2** | 87.9 | 89.6 | 91.5 | 95.8 | 73.2 | 89.6 |
| General-purpose ARCHON Architecture | 29 | 98.3 | 94.8 | 94.6 | 98.1 | 97.3 | 82.1 | 94.2 |
| Task-specific ARCHON Architectures | 34 | **98.2** | **96.7** | **95.6** | **98.5** | **98.8** | **84.2** | **95.7** |

Table 31: **MixEval Results by Sub-Dataset**: For the average computed, we do not introduce any weighting for each dataset.

| Model / LLM System | Infer. Calls | MixEval - Sub-Datasets | | | | | | |
| --- | --- | --- | --- | --- | --- | --- | --- | --- |
| | | GSM8K | TriviaQA | DROP | MATH | BBH | AGIEval | Average |
| GPT-4o - 2024-05-13 | 1 | 72.3 | 70.5 | 70.2 | 94.4 | 80.0 | 53.5 | 73.5 |
| Claude 3.5 Sonnet | 1 | 87.3 | 75.5 | 79.3 | 82.5 | 80.0 | 74.6 | 79.9 |
| Llama 3.1 405B Instruct | 1 | 98.7 | 71.2 | 70.7 | 86.9 | 78.8 | 62.0 | 78.1 |
| General-purpose ARCHON Architecture | 33 | 96.7 | 82.7 | 83.2 | 93.4 | 82.0 | 76.7 | 85.8 |
| Task-specific ARCHON Architectures | 37 | **98.9** | **86.2** | **85.2** | **96.2** | **86.0** | **80.1** | **88.8** |

Table 32: **MixEval-Hard Results by Sub-Dataset**: For the average computed, we do not introduce any weighting for each dataset.

| | GSM8K | MMLU Math | HumanEval Python | MBPP |
| --- | --- | --- | --- | --- |
| **Model** | Pass@1 | Pass@1 | Pass@1 | Pass@1 |
| GPT-4o | 97.1% | 84.8% | 89.0% | 87.5% |
| Claude 3.5 Sonnet | 96.8% | 90.9% | 90.2% | 88.9% |
| Llama 3.1 405B Instruct | 95.9% | 85.4% | 90.2% | 88.6% |

Table 33: **Additional Math and Code Benchmarks Explored**

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
