# OpenReview forum: "Archon: An Architecture Search Framework for Inference-Time Techniques"
_ICLR.cc/2025/Conference — Submitted to ICLR 2025_

### Official Review · Reviewer_w3od · 2024-11-03

**Soundness:** 3
**Presentation:** 3
**Contribution:** 2
**Rating:** 5
**Confidence:** 2

**Summary:**

ARCHON is a modular framework that addresses the challenge of designing efficient and effective inference-time architectures for large language models (LLMs). The framework focuses on optimizing various inference-time techniques, such as ensembling, ranking, fusion, critiquing, verification, and unit testing, to improve LLM performance on tasks like instruction-following, reasoning, and coding. ARCHON employs a hyperparameter optimization approach called Inference-Time Architecture Search (ITAS) using Bayesian optimization to explore the vast design space and select the most effective configuration. The paper demonstrates that ARCHON outperforms existing state-of-the-art models and inference-time architectures, achieving significant performance gains across multiple benchmarks.

**Strengths:**

## Comprehensive Framework Design:
The ARCHON framework introduces a detailed and modular architecture that provides a systematic way to apply and optimize inference-time techniques for large language models. This innovation addresses a critical gap in current methodologies, which often focus on singular techniques. ARCHON's approach is especially compelling in its structured, neural-network-inspired layering of methods, allowing for both parallel and sequential inference-time operations. The paper meticulously defines each component, such as Generators, Fusers, Rankers, Critics, and Unit Test Generators, and explains how they interact within the framework to optimize results.
## Substantial Performance Improvements:
The quantitative results are striking. ARCHON achieves an average increase of 15.1 percentage points in accuracy across benchmarks, outperforming models like GPT-4 and Claude 3.5 Sonnet. This improvement is not superficial but demonstrated over a diverse set of benchmarks, including instruction-following (e.g., MT-Bench, AlpacaEval 2.0), reasoning (e.g., MixEval, MixEval-Hard), and coding tasks (e.g., CodeContests). The use of detailed performance metrics, including Pass@1 for coding and win rates for instruction-following, adds credibility to these claims.
## Detailed Analysis of Technique Interactions:
The paper rigorously examines how different inference-time techniques complement each other. For example, the Ranker is shown to improve response quality by 5.9 percentage points when paired with the Critic, and adding multiple layers of Fusers increases the richness of responses in instruction-following tasks. These insights demonstrate a nuanced understanding of the interplay between techniques, which is often lacking in similar studies.
## Effective Use of Hyperparameter Search:
The use of Bayesian Optimization for Inference-Time Architecture Search (ITAS) is a strong point, enabling efficient exploration of a vast design space. The paper details how ITAS outperforms random and greedy search, finding optimal configurations in 96.0% of iterations while reducing the number of evaluations by over 88%. This shows a well-optimized and strategic approach to architecture search.
## Extensibility and Practical Utility:
ARCHON is designed as a plug-and-play framework, allowing practitioners to easily incorporate new LLMs and inference-time techniques. This adaptability makes it highly valuable for real-world applications, from academic research to industry use cases. The authors emphasize its potential for wide adoption by providing detailed guidelines on how to customize the framework for different tasks and compute budgets.

**Weaknesses:**

The authors themselves acknowledge these components-Verifier, for example-operate underperforming on simpler tasks, there is no clear mechanism or threshold available to disable or skip them. In their failure to do so, a weakness in the design of the framework is exposed: it could result in unnecessary computational overhead and inefficiency. Without a systematic way of assessing when the contribution of a particular component is negligible, the work in ARCHON far from completes. Again, the work has not suggested mechanisms of adaptive disabling or performance-based activation of components that would go a long way in enhancing the efficiency of the framework.

The high variability that ARCHON exhibits for different tasks suggests a major shortcoming in the way it was designed. Despite such broad generalization setup, the proposed framework generalizes poorly without significant adjustments, and therefore it is not robust. The authors lost the opportunity to embed, at least discuss how methods might enhance generalization, like meta-learning approaches or task embeddings. Because of its heavy reliance on manual tuning, the framework is far from real-world applications where task conditions cannot be predicted. The fact that this is a limitation greatly reduces the appeal and practicality of ARCHON, especially in service to those users who need to find a universally applicable solution.

This dependence of the ARCHON framework on empirical testing and the configuration of pre-filtering points to one very critical limitation-it cannot adaptively change the selection of components based on real-time performance feedback or task-specific features. This rigidity suggests inefficiency because, at any given time, ARCHON may never be optimal on every task without extensive trial-and-error tuning. For instance, any more adaptive strategy, such as reinforcement learning or meta-learning, could be investigated in order to let the system automatically adapt its configuration. Why such a strategy has not been considered is not discussed by the authors, together with the motivation of the mentioned feasibility challenges, thus leaving a wide gap in the discussion of scalability and adaptability of the paper.

**Questions:**

Latency vs. Performance Trade-Off: Given the confirmed high latency and compute cost, can ARCHON be adapted for real-time applications without sacrificing significant performance? If not, how do you foresee its practical deployment in latency-sensitive environments?
Task Generalization: While ARCHON performs well on a variety of benchmarks, there seems to be variability in how effective certain components are across different tasks. Have you considered any techniques for improving generalization, such as meta-learning approaches or task embeddings, to make component selection more robust across diverse problem sets?
Dynamic Adaptation: Your methodology for component selection heavily relies on empirical testing and pre-filtering configurations. Have you considered implementing a more adaptive or dynamic strategy that could optimize component selection in real-time based on task-specific features? If not, what are the main challenges or limitations you foresee in developing such a strategy?

---

### Official Review · Reviewer_2Lsw · 2024-11-04

**Soundness:** 3
**Presentation:** 2
**Contribution:** 3
**Rating:** 6
**Confidence:** 3

**Summary:**

This paper proposes a NAS-like approach to assembling a hodge-podge of LLM inference-time techniques for a target test benchmark. The resulting architecture, called ARCHON, is able to improve by scaling the number of LLM samples/models used as a subunit. Performance is evaluated in a number of benchmarks. The paper overall studies "meta architectures" where specialized LLMs are the unit of computing. Search is done with a classic Bayes optimization algorithm with a Gaussian process.

**Strengths:**

+ The approach performs well, both in the closed/open model settings, and can produce combinations of open models that reach close to closed model performance
+ The approach is relatively inexpensive, using approximately 40x inference calls on a single query (though some opinions may differ on whether this is inexpensive)
+  The resulting architectures found in the appendix are relatively simple, and scaling is easy for certain components

**Weaknesses:**

+ The proposed approach is not quite comparable to the baselines in terms of compute. While ARCHON uses 30+ inference calls, other LLM systems are given just one. It would be fairer to uses any single building block but scaled up (if possible) to having approximately the same inference cost, to demonstrate that the combination is truly driving the performance gains as opposed to the # of inference calls. If performance was better even when giving the baselines similar compute, this could help dampen the briefly discussed limitation of additional inference calls in the conclusion.
+ Most of the LLM components in section 3.1 do not cite previous work (only the generator does). It would be good to describe what these modules are based on, especially since this paper's contribution is centered around the combination and not the proposal of new parts.
+ No standard errors reported, and other details are light/missing (i.e. the specific search algorithm)

**Questions:**

+ Figure 5's table is somewhat confusing. While ARCHON results appear to have two sets of results (general and task specific), the other LLM systems are not categorized at all. Also, the inference calls are not close to each other---is there a reason why one cannot give the baselines a similar level of compute?
+ Almost the entirety of the search algorithm is pushed to the appendix A6, which reads like a textbook overview of classic Bayesian optimization with a Gaussian process. This section should really be edited to state what the authors do, not generically about what Bayesian optimization is. For example, Gaussian processes are described as an example of a surrogate, and examples of acquisition functions are given, but the specific one used in the paper for the given approach is not described in A6. What is the specific algorithm being used?
+ The set of rules may make architecture building with these LLM components as building blocks a bit more complex than traditional architecture search. In fact, with all of these rules, the search may end up being mostly parameter search instead of true architecture search, if the overall architecture is already mostly defined by the rules. Are there examples of different "paradigms" of combinations that arise from ARCHON? or are they all mostly the same?

---

### Official Review · Reviewer_q7H3 · 2024-11-04

**Soundness:** 2
**Presentation:** 3
**Contribution:** 2
**Rating:** 5
**Confidence:** 4

**Summary:**

This paper explores the composition and interactions of popular inference-time techniques in the field of large language models (LLMs) to better understand their relationships and performance across various tasks. It introduces a framework, ARCHON, designed to combine mainstream inference-time methods and select the optimal set of models for specific benchmarks or tasks. The proposed LLM system demonstrates significant improvements across a variety of datasets, outperforming any single LLM, including the most competitive models. The paper also provides a thorough empirical analysis of how various factors influence the outcomes.

**Strengths:**

The paper provides an insightful summary of existing inference-time techniques in the LLM field, distilling their concepts into well-structured building blocks that establish a robust paradigm for constructing LLM systems. The proposed framework, ARCHON, demonstrates significant performance improvements on common benchmarks, highlighting its effectiveness. The results from the LLM component interaction experiments are also valuable in real-world practice. Additionally, the presentation is generally clear and fluid, facilitating a coherent understanding of the paper's work.

**Weaknesses:**

While this work has involved substantial effort, I recommend rejecting the paper for the following reasons:
(1) The paper offers few novel insights into inference-time techniques, and the main conclusions drawn from the experiments are rather trivial, relying primarily on parameter tuning without theoretical justification; (2) The experimental setup is neither practical nor comprehensive enough to fully demonstrate the interaction mechanisms of mentioned methods; (3) The proposed framework is of limited practical value due to the high computational costs associated with parameter searching and inference.

The paper contributes little to the understanding of the utility or interactions of inference-time techniques. The improvements in downstream tasks through the combination of techniques, as discussed in Section 4, are predictable and have been partly addressed in prior work. Simply combining these techniques and presenting performance results does not constitute a substantial contribution. Additionally, the analysis of different utilities lacks theoretical grounding. The claimed insights, such as those in Sections 4.4 and A.2, are simply descriptive observations from comparative experiments. As seen in A.2, under the specific settings (using only 70B+ open-source models), the results are unlikely to provide generalizable rules.

Regarding the experimental setup, several points require clarification. See the question part for details.

In addition to these points, the proposed framework is significantly limited by its high computational cost. Even with parallel execution of LLMs within the same layer, the system takes five times longer to respond, not to mention the increased demand for computational resources and associated costs, as highlighted in Section A.5. Handling such a system for models of different architectures and sizes would be a great challenge to both hardware and operation-maintenance service in real-world applications. Given the flexibility of the proposed framework, it would be beneficial to consider budget and cost constraints when searching for the optimal architecture in future work. Providing a more cost-effective version of the general-purpose architecture would also enhance the framework's practicality.

**Questions:**

(1) In Figure 2, the example system architecture includes two GPT-4o generators in the first layer, yet Section 3.3 states that generators are selected from the top-K best models. Is there an inconsistency in the figure, or is there something I may have misunderstood?

(2) It is puzzling that the paper does not include a lightweight version in the experiments, i.e., exclusively using models with fewer than 8B parameters. On one hand, in practical applications, it is more common to deploy a mixture of smaller models due to resource constraints, making this approach more valuable. On the other hand, the coexistence of models of different sizes might cause the ITAS algorithm to disproportionately favor larger models, potentially disregarding the smaller ones. This tendency is also evident in Appendix A.4, where almost all architectures rely predominantly on large models. Why is there no consideration for the exclusive use of small-sized models?

(3) In Section 4.4, the construction strategy for the General-purpose ARCHON Architecture is unclear. What is meant by ``maximizing performance over all benchmarks''? Is ITAS performed sequentially on sampled subsets from each dataset, or only once on the concatenated sampled subsets?

---

### Official Review · Reviewer_YLS1 · 2024-11-10

**Soundness:** 2
**Presentation:** 2
**Contribution:** 2
**Rating:** 3
**Confidence:** 3

**Summary:**

This paper presents ARCHON, a framework for stacking different LLM components to leverage test-time compute to improve performance. While a more general architecture is introduced, the actual pipeline first generates K candidate solutions and then passes through the test through a number of fusion layers before a final verifier/unit test layer. Many hyperparameters are introduced to control the architecture and then they are optimized either with bayesian optimization or greedy search on a subset (20%) of the test set. This yields a pipeline that substantially improves performance on a variety of benchmarks like MT-bench and MATH.

**Strengths:**

1. The results seem to be generally strong. Even if it requires many calls to the models to yield the final result, the reported performance numbers on a variety of benchmarks are impressive, often matching or exceeding state of the art performance.

2. The authors produce an open sourced framework so that others can reproduce and build on the results.

**Weaknesses:**

1. There are some weird formatting issues with the paper. All the characters seem to be compressed compared to other submissions I am reviewing. But also, the bottom margins are substantially enlarged.

2. The comparisons often don't seem to be very fair and baselines are lacking. We should match the FLOP/token/dollar budgets between different methods to see which methods are most effective at turning additional compute into performance. The results are often confounded by comparing across substantially different budgets. For example, comparing elaborate inference techniques with single model calls. This is interesting to show that open source models can be augmented to out-perform a closed source model, but for good science we would want to see an apples-to-apples comparison of different inference techniques on a fixed model or models. Fixing the budget in this comparison is important. For example, in the paper in Figure 3 we see a comparison of different ablations, but some of them require much more compute than others since they add in extra model components. It would be useful to compare with a fixed budget where e.g. if we add fusers, then we need to reduce K to keep the total FLOPs of the system constant.

3. It is not clear what is the real novelty of the method or what the underlying principles of the approach are. Each component of the model alone is a standard component from the literature, and there already exist methods for combining these sorts of things and optimizing them like DsPy and mixture of agents. The main contribution seems to be setting up a particular way to parameterize a hyperparameter optimization problem, but it is not clear why this particular version was chosen. There are a variety of ablations showing that adding various components can improve performance, but there does not seem to be much insight into what is actually going on here or how the specific hyperparameter problem was chosen.

4. The presentation does not track the actual method in the experiments very well, making it somewhat confusing. The presentation in 3.2 is very broad encompassing many architectures. Then 3.3 seems to choose one very specific architecture, which all the experiments use, but it is not clear exactly what this architecture is. Perhaps it would be more clear to just present the actual method used in the experiments directly, and then discuss generalizations as future work rather than flag-planting something that is not actually attempted.

**Questions:**

See weaknesses.

---

### Meta-Review · Area_Chair_bEqa · 2024-12-21

**Metareview:**

This paper combines different LLMs and inference-time techniques to obtain better performance than with one of them alone. The problem of selecting the right combination of LLM and inference-time techniques is viewed as a hyperparameter optimization problem and addresses it with an off-the-shelf Bayesian optimization method. Reviewers highlighted various strengths, including strong empirical results, often exceeding SOTA methods, and a detailed analysis of the interplay between different methods. Criticisms included the low novelty and high compute time, combined with the lack of meta-learning. The most serious concern for me for the decision to reject was that of overselling and overblown contributions by Reviewer YLS1. While I decided for rejection following most reviewers, I do see a lot of potential in this work and encourage the authors to continue this line of work and submit to the next venue.

**Additional Comments On Reviewer Discussion:**

Various concerns were brought up and many of the them addressed. The high compute time is an issue, but the authors did provide a compute-matched evaluation. Reviewer YLS1's remarks on overblown claims and overselling were not effectively rebutted.

---

### Decision · Program_Chairs · 2025-01-22

Reject